# Genetic specification of left–right asymmetry in the diaphragm muscles and their motor innervation

**Camille Charoy[1], Sarah Dinvaut[1], Yohan Chaix[1], Laurette Morlé[1], Isabelle Sanyas[1], Muriel Bozon[1], Karine Kindbeiter[1], Bénédicte Durand[1], Jennifer M Skidmore[2,3], Lies De Groef[4], Motoaki Seki[5], Lieve Moons[4], Christiana Ruhrberg[6], James F Martin[7], Donna M Martin[2,3,8], Julien Falk[1†], Valerie Castellani[1*†]**

[1]University of Lyon, Claude Bernard University Lyon 1, INMG UMR CNRS 5310, INSERM U1217, Lyon, France; [2]Department of Pediatrics, University of Michigan Medical Center, Ann Arbor, United States; [3]Department of Communicable Diseases, University of Michigan Medical Center, Ann Arbor, United States; [4]Animal Physiology and Neurobiology Section, Department of Biology, Laboratory of Neural Circuit Development and Regeneration, Leuven, Belgium; [5]Research Center for Advanced Science and Technology, University of Tokyo, Tokyo, Japan; [6]UCL Institute of Ophthalmology, University College London, London, United Kingdom; [7]Baylor College of Medicine, Houston, United States; [8]Department of Human Genetics, University of Michigan Medical Center, Ann Arbor, United States

**\*For correspondence:** valerie. castellani@univ-lyon1.fr

[†]These authors contributed equally to this work

**Competing interests:** The authors declare that no competing interests exist.

**Abstract** The diaphragm muscle is essential for breathing in mammals. Its asymmetric elevation during contraction correlates with morphological features suggestive of inherent left–right (L/R) asymmetry. Whether this asymmetry is due to L versus R differences in the muscle or in the phrenic nerve activity is unknown. Here, we have combined the analysis of genetically modified mouse models with transcriptomic analysis to show that both the diaphragm muscle and phrenic nerves have asymmetries, which can be established independently of each other during early embryogenesis in pathway instructed by Nodal, a morphogen that also conveys asymmetry in other organs. We further found that phrenic motoneurons receive an early L/R genetic imprint, with L versus R differences both in Slit/Robo signaling and MMP2 activity and in the contribution of both pathways to establish phrenic nerve asymmetry. Our study therefore demonstrates L–R imprinting of spinal motoneurons and describes how L/R modulation of axon guidance signaling helps to match neural circuit formation to organ asymmetry.

## Introduction

The diaphragm is the main respiratory muscle of mammalian organisms, separating the thoracic and abdominal cavities. Many diseases, including congenital hernia, degenerative pathologies and spinal cord injury, affect diaphragm function and thereby cause morbidity and mortality (*Greer, 2013*; *McCool and Tzelepis, 2012*). Despite the large interest given to diaphragm function in various physiological and pathological contexts (*Lin et al., 2000*; *Misgeld et al., 2002*; *Strochlic et al., 2012*), little attention has been paid to the embryological origin of left–right (L/R) asymmetries in diaphragm morphology and contraction, in part because they were inferred to be simply an adaptation to the structure of other, surrounding asymmetric organs such as the lungs (*Laskowski et al., 1991*; *Whitelaw, 1987*). In the present study, we investigated the origin and the mechanisms responsible for the

**eLife digest** The diaphragm is a dome-shaped muscle that forms the floor of the rib cage, separating the lungs from the abdomen. As we breathe in, the diaphragm contracts. This causes the chest cavity to expand, drawing air into the lungs. A pair of nerves called the phrenic nerves carry signals from the spinal cord to the diaphragm to tell it when to contract. These nerves project from the left and right sides of the spinal cord to the left and right sides of the diaphragm respectively.

The left and right sides of the diaphragm are not entirely level, but it was not known why. To investigate, Charoy et al. studied how the diaphragm develops in mouse embryos. This revealed that the left and right phrenic nerves are not symmetrical. Neither are the muscles on each side of the diaphragm. Further investigation revealed that a genetic program that establishes other differences between the left and right sides of the embryo also gives rise to the differences between the left and right sides of the diaphragm. This program switches on different genes in the left and right phrenic nerves, which activate different molecular pathways in the left and right sides of the diaphragm muscle.

The differences between the nerves and muscles on the left and right sides of the diaphragm could explain why some muscle disorders affect only one side of the diaphragm. Similarly, they could explain why congenital hernias caused by abdominal organs pushing through the diaphragm into the chest cavity mostly affect the left side of the diaphragm. Further studies are now needed to investigate these possibilities. The techniques used by Charoy et al. to map the molecular diversity of spinal cord neurons could also lead to new strategies for repairing damage to the spinal cord following injury or disease.

establishment of the diaphragm asymmetries, including motor innervation by the left and right phrenic motoneurons that arise in the spinal cord at cervical levels C3 to C5 (*Greer et al., 1999*; *Laskowski and Owens, 1994*). Our findings show that both the diaphragm muscle and phrenic nerves have asymmetries, which are established independently of each other during early embryogenesis.

## Results

As many L/R asymmetries are determined prenatally (*Sun et al., 2005*), we analyzed the diaphragm innervation of mouse embryos on embryonic day (E) 15.5, when synaptic contacts begin to be established in this organ (*Lin et al., 2001*). We observed that the phrenic nerves split into primary dorsal and ventral branches when reaching the lateral muscles, whereby the distance from the end-plate to the nerve entry point differs between the left and right side and results in a characteristic 'T' -like pattern on the left and 'V' -like pattern on the right (*Figure 1A*; *Figure 1—figure supplement 1A, B*). Similar differences in the L/R branching patterns are present in the human diaphragm (*Hidayet et al., 1974*) (*Figure 1—figure supplement 1C*). Additionally, we observed an asymmetric number of branches defasciculating from the left and right primary nerves to innervate the motor end-plates (*Figure 1A*; *Figure 1—figure supplement 1A,B*). We further found that the L/R distribution of acetylcholine receptor (AchR) clusters at the nascent neuromuscular junctions also differed, with a $2.1 \pm 0.2$-fold increase in the medio-lateral scattering of AchR clusters on the right side of the diaphragm compared to the left side (N = 11, p<0.001 Wilcoxon) (*Figure 1B*; *Figure 1—figure supplement 2A,B*). The time course analysis revealed that these asymmetric nerve patterns arose at E12.5, concomitantly with branch formation (*Figure 1C–E*; *Figure 1—figure supplement 3A–C*). Thus, phrenic branch patterns exhibit clear asymmetries before synapse formation and fetal respiratory movements (*Lin et al., 2001*, *2008*), and are therefore unlikely to be induced by nerve activity or muscle contraction.

We therefore asked whether diaphragm nerve asymmetry was genetically hard-wired downstream of Nodal signaling, which initiates a left-restricted transcriptional cascade to establish visceral asymmetry (*Komatsu and Mishina, 2013*; *Nakamura and Hamada, 2012*). To answer this question, we examined two complementary types of mouse mutants that have defective Nodal signaling and ensuing lung isomerism. First, we examined *Pitx2*$^{\Delta C/\Delta C}$ embryos lacking PITX2C, a transcription

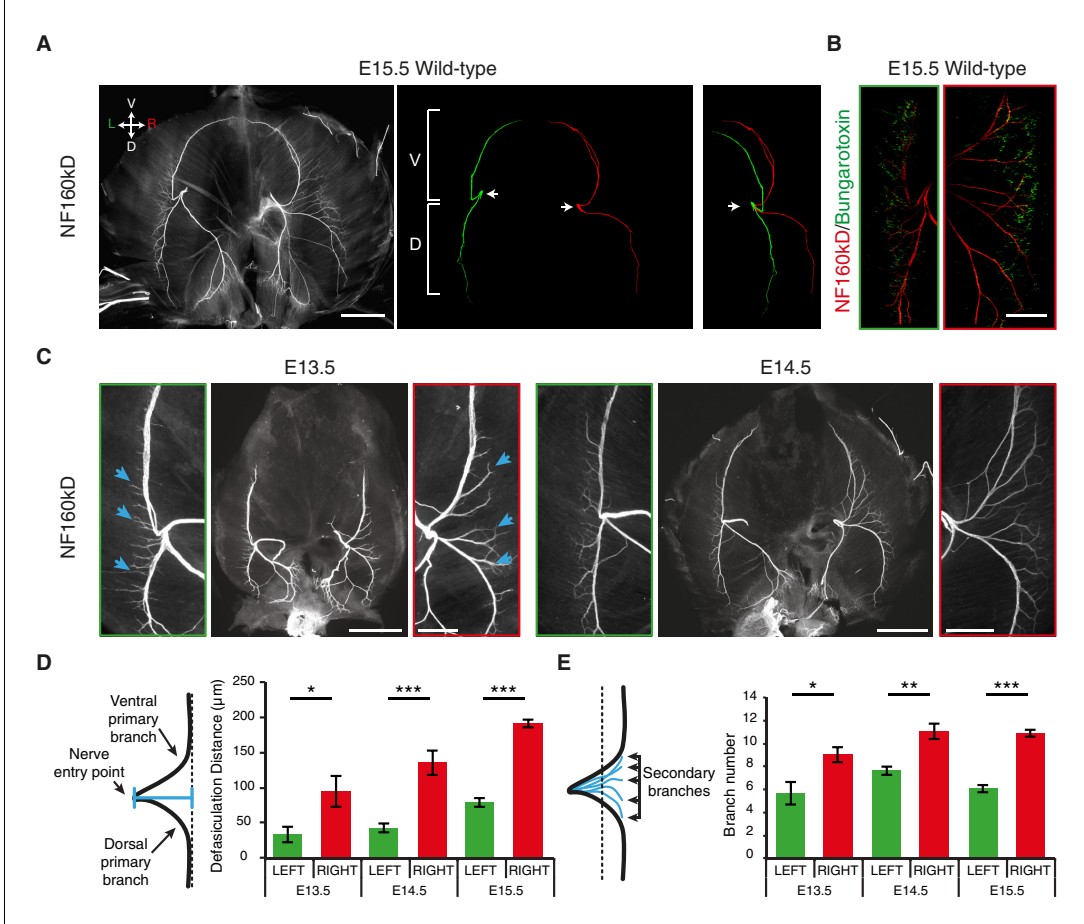

**Figure 1.** L/R asymmetries of the phrenic nerve patterns are established from the onset of diaphragm innervation. (**A**) Neurofilament (NF) staining showing the branching patterns of the left and right phrenic nerves in whole-mount E15.5 mouse diaphragm. Left and right primary branches are pseudocolored (middle panel) in green and red, respectively. (See *Figure 1—figure supplement 1A*, for complete branch traces). L/R asymmetry is especially apparent after superimposing the left and right primary branches (right panel). Arrows point to the nerve entry points. Images are top views of the whole diaphragm, oriented as indicated in the top left hand corner of the left panel (V, Ventral; D, Dorsal; L, Left; R, Right). (**B**) NF and Bungarotoxin staining showing the asymmetry of acetylcholine receptor clusters and nerve domains on the left (left panel, green frame) and right (right panel, red frame) diaphragm muscles of an E15.5 embryo (see *Figure 1—figure supplement 2* for quantification). (**C**) NF staining showing the patterns of left and right phrenic nerves at E13.5 and E14.5. Green- and red-framed panels show enlarged images of the left and right phrenic nerves, respectively. (**D**) Schematics showing the method used to quantify the defasciculation distance (shown in blue), from the nerve entry point to the dotted line and histogram of the defasciculation distance at E13.5, E14.5 and E15.5 (E13.5 — left 32.76 ± 11.01, right 94.82 ± 21.94, N = 9, p=0.0106; E14.5 — left 42.56 ± 4.16, right 135.71 ± 10.20, N = 8, p=0.00015; E15.5 — left 77.16 ± 7.32, right 188.51 ± 7.01, N = 18, p=4 E-10, Mann-Whitney). (**E**) Schematics showing the method used to quantify the secondary branch number by counting the number of NF-positive fascicles that crossed the dotted line positioned at 80% of the defasciculation distance and histogram of the secondary branch number at E13.5, E14.5 and E15.5 (E13.5 — left 5.55 ± 0.96, right 8.88 ± 0.65, N = 9, p=0.0288; E14.5 — left 7.5 ± 0.38, right 10.88 ± 0.69, N = 8, p=0.00117; E15.5 — left 5.94 ± 0.31, right 10.7 ± 0.3, N = 18, p=2.35 E-7, Mann-Whitney). Histograms show the mean ± SEM for each stage. Scale bars: 200 μm (A,C); 100 μm (B). Numerical values used to generate the graphs are accessible in *Figure 1—source data 1*.

The following source data and figure supplements are available for figure 1:

**Source data 1.** Left and right measures of the defasciculation distance and branch number in E13.5, E14.5 and E15.5 mouse embryos.

**Figure supplement 1.** Phrenic nerve patterns and quantification in mice and L/R nerve asymmetry in a human diaphragm.

**Figure supplement 2.** L/R differences of acetylcholine clusters during synaptogenesis.

**Figure supplement 2—source data 1.** Left and right endplate thicknesses measured from Bungarotoxin labeling in E15.5 mouse embryos.

*Figure 1 continued*

**Figure supplement 3.** Stereotypy and variability of L/R asymmetry of the phrenic nerve patterns.
**Figure supplement 3—source data 2.** Paired analysis of left and right defasciculation distances in E14.5 mouse embryos.

factor downstream of Nodal (*Essner et al., 2000*; *Liu et al., 2001*; *Schweickert et al., 2000*). In the absence of PITX2C, Nodal signaling is interrupted, which causes a right pulmonary isomerism (i.e. the left lung has three main lobes like the right lung, instead of only one) (*Liu et al., 2001*, *2002*). Second, we examined *Rfx3⁻/⁻* embryos lacking RFX3, which is essential for cilia function that helps to distribute Nodal to the left side of the body. As a result, some *Rfx3⁻/⁻* embryos exhibit bilateral Nodal expression and left pulmonary isomerism (i.e. the right lung has one lobe like the left lung) (*Bonnafe et al., 2004*). We found that diaphragm L/R nerve asymmetries were lost in both *Pitx2^{ΔC/ΔC}* and *Rfx3⁻/⁻* embryos with impaired visceral asymmetries at E14.5 (*Figure 2A–E*) (number of secondary branches, Wt versus mutant with lung isomerism: PITX2C, p=4.493E-5; RFX3, p=0.002884; defasciculation distance, Wt versus mutant with lung isomerism: PITX2C, p=0.001268; RFX3, p=2.719E-6, Mann-Whitney). Thus, the Nodal pathway is essential for the establishment of diaphragm nerve asymmetry.

We next asked whether phrenic nerve asymmetry has an environmental origin, because it is conceivable that the lung buds confer L/R asymmetry-inducing signals to nerves that are navigating

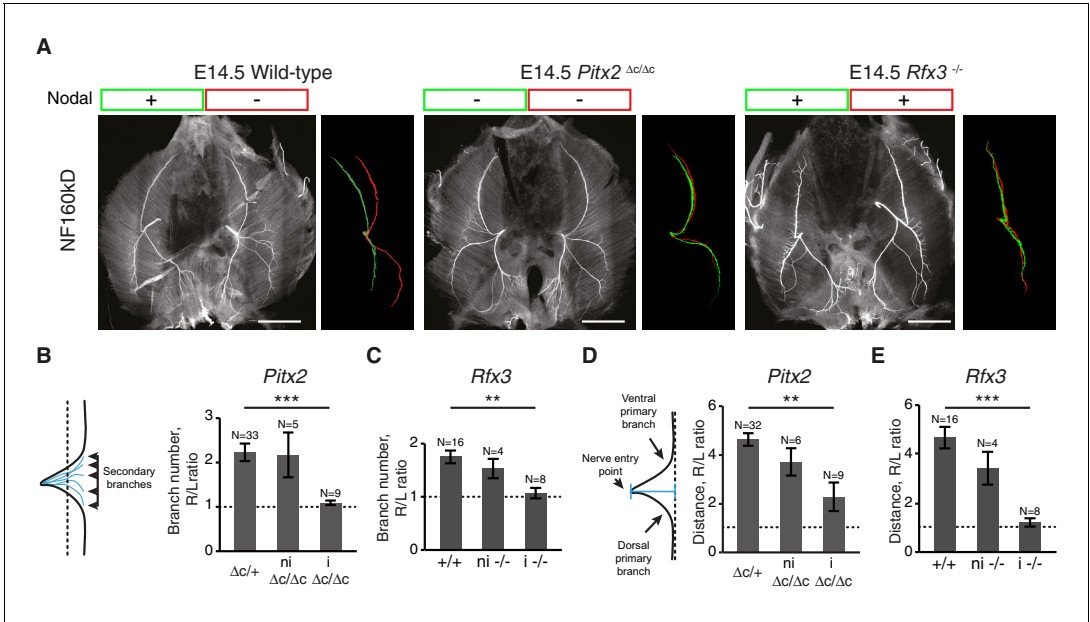

**Figure 2.** L/R asymmetries of the phrenic nerve patterns require Nodal signaling. (**A**) NF staining of E14.5 diaphragms from wild-type, *Pitx2^{ΔC/ΔC}* and *Rfx3⁻/⁻* embryos with the respective superimposed L/R nerve pattern and the Nodal expression. (**B–C**) Schematic of the secondary branches quantification and histograms of the R/L ratios of secondary branches: *Pitx2^{ΔC/+}* and *Pitx2^{+/+}* 2.23 ± 0.20, versus *Pitx2^{ΔC/ΔC}* with lung isomerism 1.09 ± 0.05, p=4.493E-5 (**B**); *Rfx3^{+/+}* and *Rfx3⁻/⁺* 1.75 ± 0.12, versus *Rfx3⁻/⁻* with lung isomerism 1.07 ± 0.10, p=0.002884, Mann-Whitney (**C**). (**D–E**) Schematic of the defasciculation distance measurements and histograms of the R/L ratios of defasciculation distance for: *Pitx2^{ΔC/+}* and Pitx2^{+/+} 4.63 ± 0.26, versus *Pitx2^{ΔC/ΔC}* with visceral isomerism: 2.28 ± 0.59, p=0.001268, Mann-Whitney (**D**); *Rfx3^{+/+}* and *Rfx3⁻/⁺* 4.62 ± 0.43, versus *Rfx3⁻/⁻* with visceral isomerism 1.35 ± 0.19, p=2.719E-6, Mann-Whitney (**E**). Note that there is no lung isomerism in wild-type embryos. Histograms show the mean ± SEM. Numbers above bars indicate the number of embryos analysed. ni, non-isomeric (embryos that did not exhibit visceral isomerism); i, isomeric. Scale bars: 200 μm. Numerical values used to generate the graphs are accessible in *Figure 2—source data 1*.
The following source data is available for figure 2:

**Source data 1.** Ratios of the defasciculation distance and branch number in E14.5 mouse embryos of *Pitx2C* and *Rfx3* lines.

close by (*Figure 3A,B*). However, the analysis of *Pitx2*$^{\Delta C/\Delta C}$ and *Rfx3*$^{-/-}$ mutants showed that the pattern of nerve asymmetry did not always correlate with the pattern of lung asymmetry; for example, in 2/10 *Pitx2*$^{\Delta C/\Delta C}$ embryos, nerve patterns were normal even though the lungs were isomerized

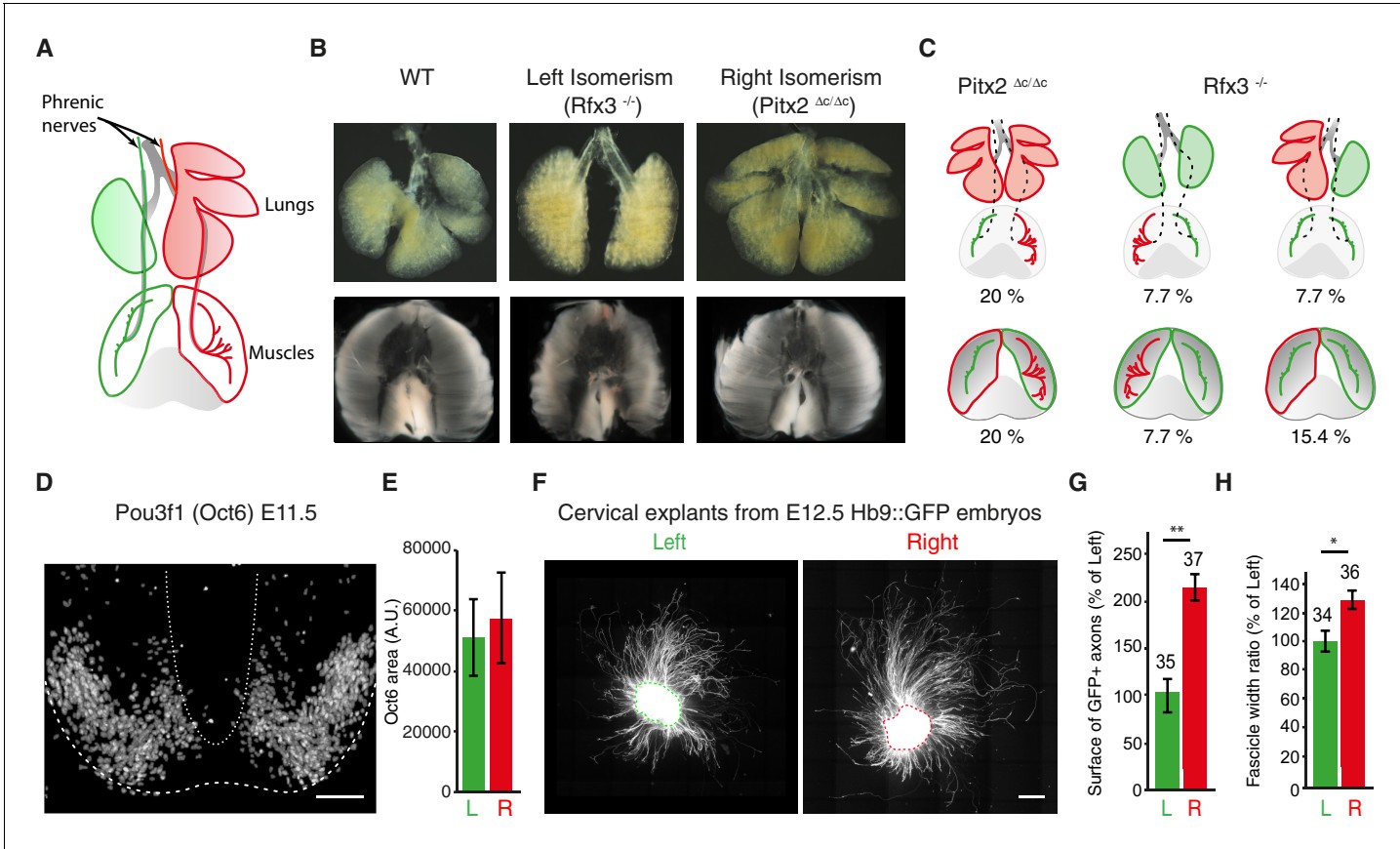

**Figure 3.** The asymmetry of phrenic circuits results from an intrinsic neuronal program. (A) Schematic representation of the organisation of the phrenic nerves as they pass through the lungs and reach the diaphragm. (B) Photomicrographs of the expected L/R asymmetry of lungs and diaphragm muscles at E14.5 in wild-type embryos and the altered L/R asymmetry observed in the *Rfx3*$^{-/-}$ and *Pitx2*$^{\Delta C/\Delta C}$ mutant embryos. Quantification of diaphragm muscle asymmetry: *Pitx2*$^{+/+}$ and *Pitx2*$^{\Delta C/+}$ 6.25 ± 0.68, N = 20, versus *Pitx2*$^{\Delta C/\Delta C}$iso 0.26 ± 0.6, N = 6; *Rfx3*$^{+/+}$ and *Rfx3*$^{-/+}$ 7.02 ± 0.74, N = 17 versus *Rfx3*$^{-/--}$iso 0.72 ± 1.6, N = 7 (see methods). (C) Schematic representation of L/R asymmetries in the lungs, diaphragm muscles and phrenic nerves. A colour code is used to show the uncoupling occurring between phrenic nerve and lung asymmetries or phrenic nerve and diaphragm muscle asymmetries. Any structure represented in green is indicative of its left characteristics, whether it is observed on the left or the right side of the embryo, whereas red structures represent right characteristics. (D) Pou3f1 (Oct6) staining showing the pool of phrenic motoneurons, projection formed by serial sections of the entire cervical region of an E11.5 spinal cord embryo. (E) Histogram showing the area positive for the Pou3f1 (Oct6) labeling in the left and right cervical motoneuron domains (N = 3, p=0.5, Wilcoxon signed rank). (F) GFP staining of ventral cervical spinal cord explants from E12.5 HB9:: GFP embryos; the dashed line is indicative of the explant border. (G) Quantification of the area occupied by GFP-positive axons for left and right explants (left — 100% ± 17.4; right — 214% ± 30.2, p=0.0045, Mann-Whitney). (H) Quantification of the width ratio (see *Figure 3—figure supplement 1* for quantification details) (left —100% ± 7.3; right — 127% ± 8.0, p=0.0127, Mann-Whitney). Numbers above bars indicate the numbers of explants analysed. Histograms show the mean ± SEM. Scale bars: 100 µm (D), 200 µm (F). Numerical values used to generate the graphs are accessible in *Figure 3—source data 1*.

The following source data and figure supplements are available for figure 3:

**Source data 1.** Pool size and in vitro axon growth from left and right motoneurons.
**Figure supplement 1.** Uncoupling between lung or muscle and nerve asymmetry and intrinsic L/R differences of axon growth from cultured cervical motoneuron explants.
**Figure supplement 1—source data 1.** Distribution of defasciculation ratios in the *Pitx2C* mouse line.

(20%; *Figure 3C*; *Figure 3—figure supplement 1A,B*). Moreover, 1/13 *Rfx3$^{-/-}$* embryos exhibited nerve isomerism together with pulmonary *situs inversus*, and nerve patterns were reversed in 1/13 embryo with lung isomerism (7.7% and 7.7%; *Figure 3C*; *Figure 3—figure supplement 1A,B*). Alternatively, it is conceivable that muscle asymmetry controls nerve asymmetry. In agreement with this possibility, L/R asymmetry of the lateral diaphragm muscles was lost in both *Pitx2$^{\Delta C/\Delta C}$* and *Rfx3$^{-/-}$* mutants (*Figure 3B*). However, muscle width did not correlate with changes in nerve patterns in 2/10 *Pitx2$^{\Delta C/\Delta C}$* embryos or in 6/13 *Rfx3$^{-/-}$* embryos (20% and 46.2%, respectively). For example, muscle isomerism could be observed in 1/13 *Rfx3$^{-/-}$* embryos that have normal nerve patterns (7.7%) or in 1/13 *Rfx3$^{-/-}$* embryos with reversed nerve patterns (7.7%). Finally, nerves were isomerized in 2/13 *Rfx3$^{-/-}$* embryos that exhibit normal L/R muscle asymmetry (15.4%) (*Figure 3C*; *Figure 3—figure supplement 1C*). Together, these findings raise the possibility that phrenic motoneurons possess intrinsic L/R differences that are established independently of visceral and muscle asymmetries.

3D reconstructions of cervical spinal cord tissue immunolabeled with Pou3f1/Oct6, whose expression has been reported in motoneurons (*Philippidou et al., 2012*), did not reveal any obvious differences in the L/R organization of the cervical motoneuron pools in the spinal cord (*Figure 3D–E*). We therefore explanted phrenic motoneuron-enriched Hb9::GFP spinal cord tissue (*Wichterle et al., 2002*) to follow the behavior of motor axons as they extended from the explants independently of the surrounding organs (*Figure 3—figure supplement 1D*). We observed that axons explanted from right tissue extended over longer distances and were organized differently than axons explanted from left tissue (*Figure 3F–H*; *Figure 3—figure supplement 1E–F*). This observation suggests that intrinsic factors present within the ventral spinal cord confer different behaviors to left and right motoneuron axons.

To identify molecular determinants of L/R differences in phrenic axon growth, we laser-captured left versus right GFP-positive cervical motoneurons from Hb9::GFP transgenic E11 embryos for microarray analysis (*Figure 4A*). The presence of several markers for phrenic motoneurons (e.g. Pou3f1/Oct6, Islet1 and ALCAM) in the microarray data demonstrated the accuracy of the dissection procedure (*Figure 4—figure supplement 1A–B*). Consistent with the lack of obvious anatomical differences distinguishing left and right Pou3f1/Oct6$^{+}$ cervical motoneuron populations, none of these markers had asymmetric expression levels. We further observed that amongst 22,600 transcripts expressed above background, 146 were enriched on the left and 194 on the right, with a predominance of transcripts encoding nuclear proteins (differentially enriched transcripts: right 35.56% versus left 26.02%; *Figure 4B*; *Figure 4—source data 1* and *2*). Immunoblotting confirmed that Morf4l1, a protein involved in histone acetylation/deacetylation and chromatin remodeling and reported to be essential for neural precursor proliferation and differentiation (*Chen et al., 2009*; *Boije et al., 2013*), was enriched in the left cervical motoneuron domain (L/R fold-change 1.81 ± 0.163, p=0.0022, Mann-Whitney; *Figure 4C–E*). Xrn2, a protein regulating RNA processing and miR stability that regulates miR expression in neurons (*Kinjo et al., 2013*), was also enriched in the left cervical motoneuron domain (L/R fold-change 1.37 ± 0.13, p=0.028; Mann-Whitney; *Figure 4—figure supplement 1C*). Thus, cervical motoneurons are intrinsically L/R-specified.

To determine whether molecular differences in L/R specification manifest themselves in differential axon guidance responses, we studied mice lacking Slit/Robo signaling, which is known to regulate the fasciculation of phrenic axons (*Jaworski and Tessier-Lavigne, 2012*). In agreement with prior reports, we observed defective nerve defasciculation in *Robo1$^{-/-}$;Robo2$^{-/-}$* double mutants (*Figure 5A*). Notably, defasciculation of the left nerve was as high as that of the right nerve and assumed a similar pattern in the left and right diaphragm, rather than adopting the normal asymmetric pattern seen in wild-type littermates (*Figure 5A*). Partial symmetrization was observed in double heterozygous mutants, indicating concentration-dependent sensitivity of phrenic nerve axons to Slit signals (*Figure 5A*).

To determine whether differential levels of *Slit/Robo* signaling dictate the L/R pattern of phrenic nerve fasciculation, we examined their transcript levels, but found no evidence for lateralized expression of the transcripts for *Robo1*, the major regulator of diaphragm innervation (*Jaworski and Tessier-Lavigne, 2012*), or the ligands of Robo1: Slit1, Slit2 and Slit3 (*Figure 5—figure supplement 1A–B*). By contrast, we identified L/R differences in Robo1 protein by immunoblotting of phrenic motor neuron-enriched cervical spinal cord tissue. Robo1 was detected in one long and two short forms (*Figure 5—figure supplement 1C*), whereby the long Robo1 form migrating as a 250 kDa protein was enriched in the left samples and the short forms migrating as 120 kDa and 130 kDa proteins

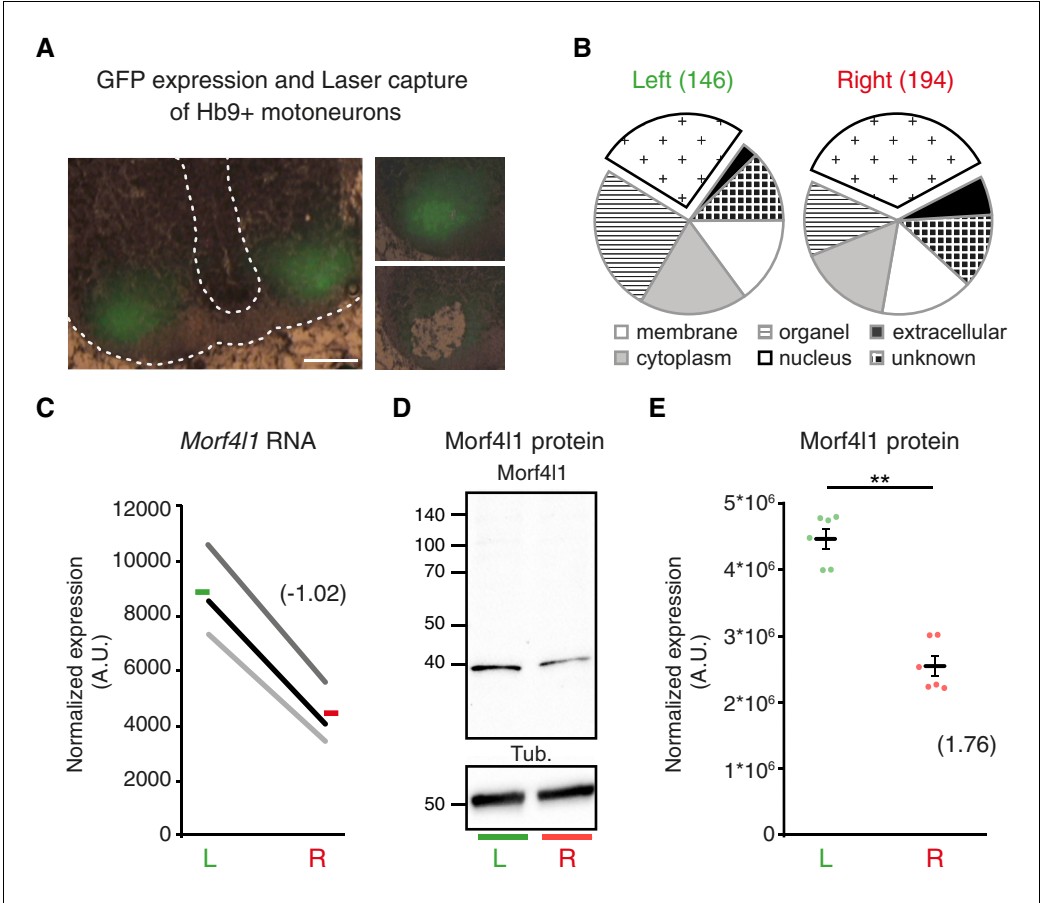

**Figure 4.** L/R molecular signature of cervical motoneurons. (**A**) Transverse sections of E11.5 Hb9::GFP embryo cervical spinal cord, illustrating the areas used for laser-capture microdissection. (**B**) Pie charts showing the proportion of left-enriched and right-enriched genes according to their Gene Ontology 'cellular component' terms. The 'nucleus' component is detached from the pie. (**C**) Ladder graph showing the left and right expression of *Morf4l1* in three embryos. Average L/R fold-change shown in brackets. (**D**) Immunodetection of Morf4l1 and loading control tubulin (Tub.) in left and right ventral cervical spinal cord tissues. (**E**) Graph showing normalized protein levels of Morf4l1 in left and right ventral cervical spinal cords from E11.5 mouse embryos. Individual values observed for the six western-blots (dots) and mean ± SEM are represented (L/R ratio: 1.81 ± 0.163, L versus R; p=0.0022, Wilcoxon signed rank). Average L/R fold-change shown in brackets. Scale bar: 100 μm. Numerical values used to generate the graphs are accessible in *Figure 4—source data 3*.

The following source data and figure supplements are available for figure 4:

**Source data 1.** List of enriched genes in the left cervical motor neurons of *HB9::GFP* embryos at E11.5.

**Source data 2.** List of enriched genes in the right cervical motor neurons of *HB9::GFP* embryos at E11.5.

**Source data 3.** Lateralization expression of Morf4l1 in cervical motoneurons.

**Figure supplement 1.** Symmetric expression of phrenic motoneuron markers, and lateralized Xrn2 expression.

**Figure supplement 1—source data 1.** RNA level of motoneuron markers and asymmetric expression of Xrn2.

were enriched in the right samples (R/L ratio— 1.22 ± 0.1, p=0.01587; Mann-Whitney, *Figure 5B*, *Figure 5—figure supplement 1D*). Even though 12 alternatively spliced isoforms have been predicted for mouse Robo1, the predicted changes in protein sequence are unlikely to account for the short forms we observed in our immunoblots, because they are predicted to change the molecular

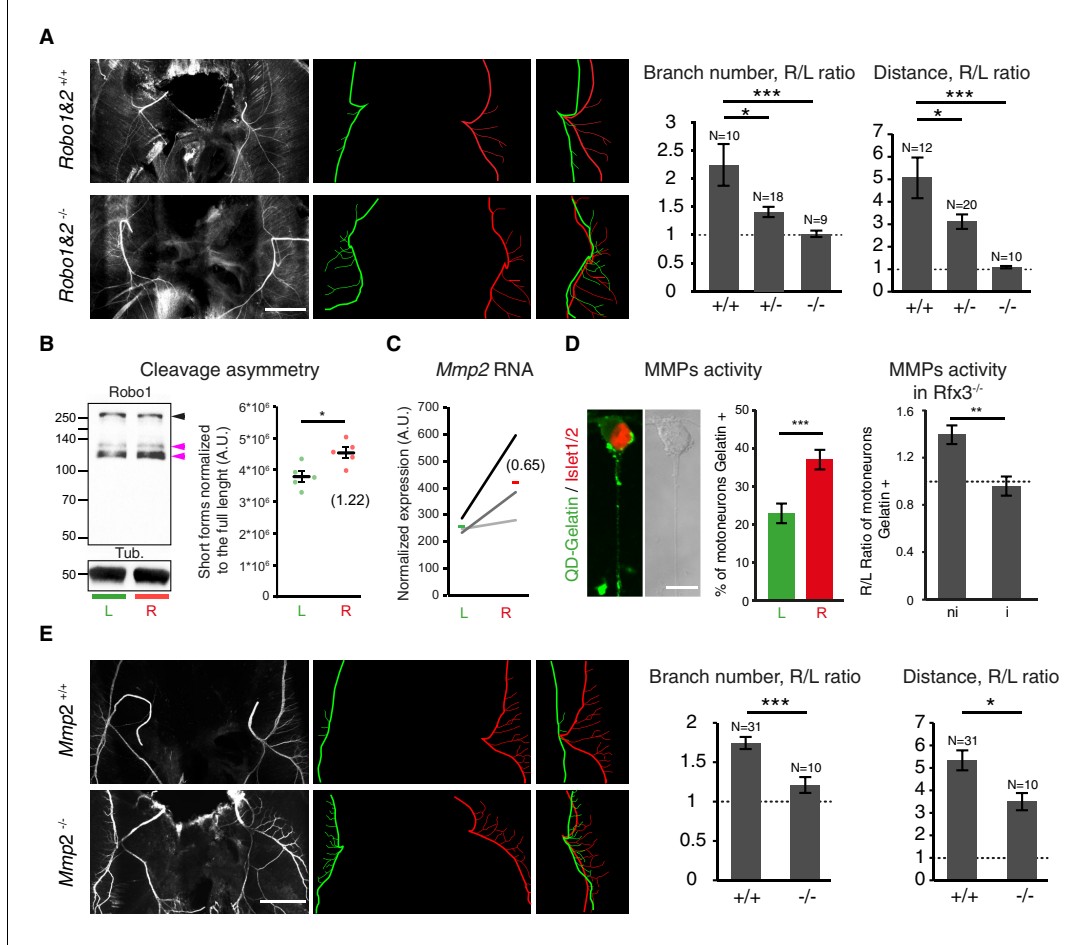

**Figure 5.** Slit/Robo signalling and MMP2 control asymmetry of L/R phrenic nerves. (**A**) NF staining of E14.5 diaphragm from *Robo1*[+/+] and *Robo2*[+/+] and *Robo1*[−/−] and *Robo2*[−/−] embryos, left and right primary branches are pseudocolored in green and red,respectively, and superimposed to show the lack of asymmetry in the *Robo*1 and 2[−/−] embryos. Histogram showing the branch number and the defasciculation distance in *Robo1*[+/+] and Robo2[+/+], *Robo1*[+/−] and *Robo2*[+/−] and *Robo1*[−/−] and *Robo2*[−/−] embryos (R/L branch ratio: *Robo1*[+/+] and *Robo2*[+/+] 2.30 ± 0.37, versus *Robo1*[−/−] and *Robo2*[−/−] 1.06 ± 0.06; p=0.00048; R/L distance ratio: *Robo1*[+/+] and Robo2[+/+] 4.99 ± 0.89, versus *Robo1*[−/−] and *Robo2*[−/−] 1.05 ± 0.07; p=3E-6, Mann-Whitney). (**B**) Immunodetection of Robo1 and loading control (Tub) in left and right HB9::GFP ventral cervical spinal cord and distribution of the relative amount of the two shorter forms (pink arrowheads) to the full-length form (black arrowhead). The graph shows the normalized left and right values obtained for the five western-blots (dots, 6–8 embryos per sample) and the mean ± SEM (R versus L: p=0.01587, Wilcoxon singed rank); average fold-change is shown in brackets (1.22 ± 0.10). Normalization between lines was done on the Robo1 long form. (**C**) Ladder graph showing the left and right expression of *Mmp2* detected by microarray in three embryos. Average Log2(R/L ratio) shown in brackets. (**D**) Photomicrograph of cultured ventral cervical spinal cord motoneuron. The combination of in situ zymmography with DQ-Gelatin and Islet1/2 staining enables the identification of motoneuron with MMP gelatinase activity. Histogram showing the amount of motoneuron with gelatinase activity in left and right samples (left 23.37% ± 2.7, N = 792 versus right 37.94% ± 2.1, N = 797; p=0.00109, Mann-Whitney). Histogram showing the gelatinase activity measured in cultures from *Rfx3*[−/−] embryos with symmetric lungs (Iso) and in cultures from *Rfx3*[+/+], *Rfx3*[+/−] embryos (*Rfx3* wt: — 1.4 ± 0.08; *Rfx3* iso — 0.96 ± 0.08, p=0.0013, Mann-Whitney). (**E**) NF staining of E14.5 diaphragms from wild-type and *Mmp2*[−/−] embryos. Left (green) and right (red) primary and secondary branch traces shown in the middle panel are superimposed in the right panel to compare the left and right patterns. Histograms showing the R/L ratios of branch number and defasciculation distances. Ratio of secondary branches: *Mmp2*[+/+] and *Mmp2*[−/+] 1.74 ± 0.07, versus *Mmp2*[−/−] 1.21 ± 0.10; p=0.00029; defasciculation distance: *Mmp2*[+/+] and *Mmp2*[−/+] 5.33 ± 0.44, versus *Mmp2*[−/−] 3.49 ± 0.38; p=0.022, Mann-Whitney. Scale bar: 200 μm (**A,E**), 10 μm (**D**). Numerical values used to generate the graphs are accessible in *Figure 5—source data 1*.

The following source data and figure supplements are available for figure 5:

**Source data 1.** Slit/Robo signalling controls asymmetry of L/R phrenic nerves and Robo1 exhibits different processing levels in left and right cervical motoneurons.

**Figure supplement 1.** Post-translational regulation of Robo1.

*Figure 5 continued*

**Figure supplement 1—source data 1.** Post-translational regulation of Robo1 and biased expression of *Mmp2*.

**Figure supplement 2.** Asymmetric expression of MMP2.

**Figure supplement 2—source data 2.** Asymmetric expression of *Mmp2* in cervical motoneurons and expression of other MMPs.

weight by just 7.1 kDa. However, both human and *drosophila* Robo1 have been shown to be processed by metalloproteases (*Seki et al., 2010*; *Coleman et al., 2010*), and potential cleavage fragments have been reported in mouse brain tissues (*Clark et al., 2002*). These findings raise the possibility that differential post-translation processing of Robo proteins may be involved in creating L/R asymmetries in diaphragm innervations.

Next, we investigated whether axon guidance effectors that were revealed by our transcriptomic analysis to exhibit asymmetric expression levels could also contribute to the L/R phrenic nerve patterns. Given that metalloproteases have emerged as important regulators of axonal behaviors during development and regeneration (*Bai and Pfaff, 2011*; *Łukaszewicz-Zając et al., 2014*; *Small and Crawford, 2016*; *Verslegers et al., 2013b*), we concentrated on these effectors. Consistent with previous expression data (GSE41013) (*Philippidou et al., 2012*), our transcriptome analysis indicated that cervical motoneurons express several metalloproteases. Interestingly, among the 7 Mmps and 13 ADAMs expressed by cervical motoneurons, Mmp2 and ADAM17 were expressed at higher levels in the right motoneurons. We focused on MMP2 because it was shown to control axon development in mouse and motor axon fasciculation in drosophila (*Miller et al., 2011*; *Gaublomme et al., 2014*; *Zuo et al., 1998*; *Miller et al., 2008*).

The microarray analysis showed that *Mmp2* transcripts were enriched in the Hb9-positive right motoneurons, which was confirmed using qRT-PCR and quantitative in situ hybridization (log2(R/L) Embryo 1 — 0.22 ± 0.08; Embryo 2 — 0.63 ± 0.11, RNAscope) (*Figure 5C*; *Figure 5—figure supplement 2A–D*). Moreover, in situ zymography with DQ-Gelatin (*Hill et al., 2012*), which is effectively cleaved by MMP2 (*Snoek-van Beurden and Von den Hoff, 2005*), showed that gelatinase activity on the axon shaft and growth cones was 1.6 times higher in right than in left motoneuron cultures (*Figure 5D*; *Figure 5—figure supplement 2E*). Remarkably, this difference was absent in motoneuron cultures prepared from $Rfx3^{-/-}$ embryos with phenotypic left isomerism (*Figure 5D*). These results provide evidence that the differential L/R MMP activity is controlled by the Nodal pathway and further suggest that MMP2 contributes to the establishment of phrenic nerve asymmetry.

We therefore analyzed the diaphragm nerve patterns in $Mmp2^{-/-}$ mice (*Itoh et al., 1997*). At E14.5, $Mmp2^{-/-}$ embryos exhibited normal lung asymmetry and well-developed phrenic branches on both sides (*Figure 5E*). Interestingly, we observed a partial symmetrization of the phrenic branches, with a right pattern that resembled the one observed on the left in control littermates in E14.5 $Mmp2^{-/-}$ embryos (*Figure 5E*). Thus, higher right MMP2 activity could contribute to promote the right pattern of phrenic nerve defasciculation.

## Discussion

Taken together, our work shows that the first asymmetry instruction in diaphragm patterning is provided by early Nodal signaling, which sets the L/R axis and visceral asymmetry of the embryo. Beyond this early mechanism, phrenic motoneurons have an intrinsic, genetically encoded L/R asymmetry that manifests itself in the differential activation of molecules that have key roles in axon guidance, including Robo1 and MMP2.

Future work should aim to address how and at which stage phrenic motoneurons are imprinted. For example, an early Nodal signal might be propagated from the lateral plate mesoderm (LPM) to the cervical spinal cord. In agreement with this idea, it has been suggested that Lefty expression in the prospective floor plate of the neural tube prevents Nodal diffusion to the left LPM (*Shiratori and Hamada, 2006*). Moreover, Lefty expression is confined to the left prospective floor plate and is reversed or expanded bilaterally in 'iv' or 'inv' mutants, which exhibit reverse visceral asymmetry (*Meno et al., 1997*). Given the key role of the floor plate in the patterning and

specification of spinal cord neuronal lineages (*Goulding et al., 1993*; *Placzek et al., 1991*), left and right floor plate cells might also differently imprint left and right spinal cord. Alternatively, or additionally, endothelial cells invading the ventral spinal cord could convey early Nodal signaling from the LPM to the spinal cord. Indeed, these cells exhibit L/R asymmetries that can be preserved during their migration (*Chi et al., 2003*; *Klessinger and Christ, 1996*). The resulting L/R imprints could occur early on during neurogenesis or later on during motoneuron differentiation. The two hypothesizes might not be exclusive. Indeed, recent work in zebrafish habenula suggests that differences in both the timing of neurogenesis and exposure to lateralized signal during neuron differentiation act in parallel to set L/R asymmetries (*Hüsken et al., 2014*). Interestingly, early imprinting of progenitors in *Caenorhabditis elegans* induces an epigenetic mark for L/R identity that drives differential genetic programs during neuron differentiation (*Cochella and Hobert, 2012*; *O'Meara et al., 2010*).

Our work provides evidence to show that a L/R imprint confers specific axon behaviors to the left and right phrenic motoneurons. For example, we found that Slit/Robo signaling is required for the establishment of asymmetric nerve patterns, which suggests that left and right phrenic motoneurons have different Slit/Robo signaling levels. Interestingly, Slit/Robo signaling has previously been shown to control phrenic axon fasciculation (*Jaworski and Tessier-Lavigne, 2012*). Consistent with an intrinsic control of Slit/Robo signaling as the cause of this axonal asymmetry, we discovered L/R differences in Robo1 protein in motoneurons, which may arise through differential proteolysis and may help to modulate responsiveness to Slit signaling, even though *Slit* and *Robo* genes are expressed similarly in the left and right motoneuron pools. In support of the idea that differential proteolysis contributes to the emergence of different Robo1 forms in the left and right phrenic motoneuron pools, Robo processing has previously been reported in other contexts (*Seki et al., 2010*; *Coleman et al., 2010*). This Robo1 processing could have different outcomes on Slit/Robo signaling. In drosophila, cleavage of Robo by ADAM10 is required for recruitment of downstream signaling molecules and the axon guidance response (*Coleman et al., 2010*). Metalloproteases can also decrease the amount of available receptors and/or terminate adhesion and signaling (*Bai and Pfaff, 2011*; *Hinkle et al., 2006*; *Romi et al., 2014*; *Hattori et al., 2000*; *Gatto et al., 2014*).

Slit/Robo signaling can control many different aspects of axon development, such as axon growth, branching, guidance or fasciculation. As primary and secondary branches are formed by selective defasciculation and because Slit/Robo signaling controls phrenic axon fasciculation, our interpretation is that the different Slit/Robo signaling abilities of left and right phrenic axons result in different axon–axon fasciculation states, with right axons having greater defasciculation behavior than the left ones. Alternatively, the Slit/Robo pathway may differentially regulate axon and branch growth, or branch trajectories, as it does for other systems of neuronal projections (*Wang et al., 1999*; *Brose et al., 1999*; *Blockus and Chédotal, 2016*). These ideas have to be taken cautiously. Differential Robo forms were assessed from spinal cord tissue essentially containing neuronal soma, and not peripheral phrenic axons. Furthermore, the tissue samples, although enriched in phrenic motoneurons by the procedure, contained additional neuronal sub-types. Further investigations are thus needed to assess with more specific tools Robo protein dynamics and distribution along phrenic axons and in the growth cones. This work will provide a better characterization of the functional outcome determined by the balance of short and long Robo forms in the establishment of phrenic nerve patterns.

Asymmetries in several genes implicated in axon guidance were observed in our transcriptome analysis. In particular, we found differences in the expression of regulators of guidance receptors activities, such as metalloproteases. Mmp2 expression level and gelatinase activity were found to be higher in right cervical motoneurons. Moreover, differential gelatinase activity between left and right motoneurons was lost in cultures from $Rfx3^{-/-}$ mutants with symmetrical Nodal signaling, suggesting that early Nodal signaling impacts on gelatinase activity in motoneurons. Mmp2 genetic loss reduced the asymmetry of the diaphragm branch pattern, suggesting that asymmetric expression of Mmp2 in motoneurons contributes to set phrenic nerve patterns.

However, in contrast to embryos lacking Pitx2 and Rfx3, embryos lacking Mmp2 only exhibited a partial symmetrization of the phrenic nerve branches. Rfx3 and Pitx2C transcription factors act at the onset of the left–right imprinting, and their genetic loss is therefore expected to abolish the entire program of L/R nerve asymmetry. By contrast, the subsequent construction of individual neuronal circuits relies on the concerted action of many different signaling pathways, whereby loss of a single pathway is not expected to disrupt the entire asymmetry program. The partial defect may be due to

the presence of other effectors of the Nodal pathway that contribute to L/R nerve asymmetries independently of MMP processing, to the co-expression of several MMPs acting with partial redundancies with each other (*Prudova et al., 2010*; *Kukreja et al., 2015*) and to the fact that MMPs have many different substrates with potentially opposite effects on the same biological process. For example, proteomic studies have identified more than 40 secreted and transmembrane substrates for MMP2 (*Dean and Overall, 2007*), of which we found 32 to be expressed in cervical motoneurons including Adam17, which is enriched in right motoneurons (*Figure 5—figure supplement 2F*, *Figure 4—source data 2*).

The MMP substrates that are responsible for asymmetric phrenic nerve patterning remain to be determined, but Slit/Robo signaling appears to be an obvious candidate. First, cleavage of human Robo1 has been suggested to be MMP-dependent, although in drosophila, Robo1 is cleaved by Adam10/Kuzbanian (*Coleman et al., 2010*; *Seki et al., 2010*). Second, short forms of Robo1, lprobably generated by proteolysis, are enriched in right motoneurons, in which MMP activity is the highest. In support, incubation of cervical spinal cord tissue with active MMP2 significantly increased the short Robo1 forms (fold change: 1.60 ± 0.23, p=0.00285, Mann-Whitney, four independent western blots, *Supplementary file 1*). Nevertheless, the L/R ratio of Robo protein forms in cervical motoneuron tissue collected from *Mmp2* null embryos, although showing a tendency towards reduction, was not statistically different from the wild-type ratio (WT: 1.22 ± 0.10, N = 5; $Mmp2^{-/-}$: 1.14 ± 0.01, N = 3; p=0.78, Mann-Whitney; *Supplementary file 2*). This might be due to an insufficient number of tested embryos. Alternatively, because short Robo1 forms were still detected, this L/R ratio might rather reflect the activity of other proteases, either compensating for MMP2 loss or also contributing to Robo processing.

An additional MMP candidate is NCAM, which is highly expressed by developing phrenic axons, controls axon-axon fasciculation, and is cleaved by MMPs (*Dean and Overall, 2007*; *Hinkle et al., 2006*). In the light of MMP redundancy and the possible involvement of other proteases in the processing of axon guidance receptors and their ligands, the *in vivo* assessment of these hypotheses will be challenging.

Finally, the genetic program for L/R identity in spinal cord motoneurons that we have described here may provide important insights into motoneuron development and diseases. For example, the L/R imprinting of spinal motoneuron might also explain why right-sided fetal forelimb movements are far more frequent than left-sided movements at developmental stages when motoneurons have not yet received any input from higher brain centers (*Hepper et al., 1998*). In addition, our description of early events controlling diaphragm formation may have broad implications for our understanding of several human conditions. Examples include congenital hernias, which generally affect the left hemi-diaphragm and can cause perinatal lethality (*Pober, 2008*), and some types of congenital myopathies that impair diaphragm function only on one side (*Grogan et al., 2005*). Our data thus provide a novel basis for investigations of molecular diversity in spinal cord neurons and for functional studies of diaphragm physiology and pathology.

## Materials and methods

### Genotyping of mouse lines

This work was conducted in accordance with the ethical rules of the European community and French ethical guidelines. Genotyping of transgenic mouse lines was performed as described in *Liu et al. (2001)* for $Pitx2^{\Delta C}$ (original line: RRID:MGI:3054744), in *Bonnafe et al. (2004)* for *Rfx3* (RRID:MGI:3045845), in *Delloye-Bourgeois et al. (2015)* for *Robo1* and *Robo2* (RRID:MGI:5522691), in *Verslegers et al. (2013b)* for *Mmp2* (RRID:MGI:3577310) and in *Huber et al. (2005)* for the HB9::GFP (RRID:IMSR_JAX:005029).

### Diaphragm immunolabeling

Diaphragms were dissected from embryos fixed overnight in 4% paraformaldehyde. After permeabilization and blocking in PBS with 5% BSA with 0.5% Triton X-100, diaphragms were incubated overnight at room temperature with the primary antibody, Neurofilament 160 kDa (1/100, RMO-270, Invitrogen, France; RRID:AB_2315286). Diaphragms were then incubated with the secondary antibody, α-mouse Alexa-555 (1/400, Invitrogen, France) with or without Alexa488-coupled α-BTX (1/50,

Molecular probes, ThermoFischer Scientific, France; RRID:AB_2313931), for 4 hr at room temperature in blocking solution. The procedure was performed entirely on freely floating diaphragms. Diaphragm imaging was then performed under an inverted microscope and a montage was constructed using the metamorph software (Molecular device, Sunnyvale, CA).

## Immunofluorescent labeling and in situ hybridization

Cryosections (20 μm) were obtained from embryos fixed overnight in 4% paraformaldehyde, embedded in 7.5% gelatin with 15% sucrose. For immunolabeling, embryonic sections or cultured neurons were incubated overnight at 4°C with Oct6 antibody (1/50; Santa Cruz, Germany) and then for 2 hr at room temperature with anti-goat secondary antibody, Alexa-488 (1/400; Invitrogen, France). Nuclei were stained with bisbenzimide (Promega, Madison, WI). In situ hybridization was performed as described previously (*Moret et al., 2007*). The probes were synthetized from the *Mmp2* IMAGE-clone plasmid (n6813184). *Mmp2 in situ* hybridization and Pou3f1 (Oct6) immunolabeling were performed on adjacent sections because the antibody could no longer recognize the Oct6 epitope after in situ hybridization.

## Images processing and quantifications

Serial Pou3f1/Oct6-labeled sections were imaged using a confocal microscope. Series of images were converted into a single stack using the ImageJ plugin Stack Builder. Images were aligned manually using morphological structures and labeled nuclei were extracted. A three-dimensional reconstruction of the Pou3f1 (Oct6) labeling from the cervical to the brachial part of the embryos was then generated in ImageJ (3D Project command). All quantifications were done using ImageJ. For quantification of defasciculation distance, we first traced the tangential straight line of the endplate (*Figure 1—figure supplement 1B*). We then traced a perpendicular line to the tangent that goes through the nerve entry point. Finally, we measured the distance from the entry point to the intersection of both lines. For branch number quantification, we traced a parallel to the tangential straight line of the endplate. The line was placed at a distance of one quarter of the defasciculation distance. We then counted the number of secondary branches that crossed the line. The endplate thickness was evaluated from the α-Btx staining. The α-Btx-positive region was outlined and divided into 30 rectangles. The average width of the rectangles was calculated. Width evaluation of endplate from the plot profile of α-Btx staining gave similar values.

## Western blot

Cervical ventral spinal cords were dissected from E11.5 HB9::GFP embryos in cold HBSS with 6% glucose (as shown in *Figure 4—figure supplement 1D*) and directly frozen in dry-ice cooled eppendorf tubes. Typically, left and right dissected tissues from 6–8 embryos were pooled and lysed in RIPA buffer with protease inhibitors for 30 min at 4°C. Western blots were performed using primary antibody (Anti-Morf4l1 (1:1000, Abcam, France – ab183663), anti-Xrn2 (1:1000, Abcam, France – ab72181, RRID:AB_2241927), anti-Robo1 (1:500 [*Seki et al., 2010*]) and secondary antibody (Anti-goat or -mouse HRP [A5420 and A4416, Sigma-Aldrich, France] at 1/5000). Image quantification was done with Image Lab4.0 software (Bio-Rad, France). Left and right data were normalized to the tubulin level for Morf4l1 and Xrn2 and to Robo1 full-length or tubulin for Robo1 short forms. To allow comparison between replicates left and right values were then normalized to have the same left plus right sum for all western blots.

## Motoneuron explant culture

E12.5 GFP-positive mouse embryos (4–6 per experiment) from HB9::GFP transgenic mice were selected and dissected using the fluorescence GFP-positive pool. Ventral cervical spinal cords were isolated (left and right parts separated) and cut into explants. Explants were cultured as described in *Moret et al. (2007)*. Immunohistochemistry was performed using Anti-Tuj1 (1:100, Millipore, France – MAB1637, RRID:AB_2210524) and anti-GFP (1:100, Invitrogen, France – A11122, RRID:AB_221569). Axon outgrowth was calculated using the ImageJ plugin NeuriteJ (*Torres-Espín et al., 2014*), which creates regions of interest (ROI) corresponding to radial concentric rings separated by 25 pixels. NeuriteJ extracted the signal from GFP-positive axons and measured the labeled surfaces between two ROIs. To quantify the total area occupied by GFP-positive axon, we summed the

surface of all ROIs. To calculate the proximo-distal index, the width of the labeled axons was calculated in the second ROI (proximal ring) and in the ROI at 30% of the maximal distance of growth (distal ring) (see *Figure 4—figure supplement 1F*). The index was calculated by dividing the width of the proximal fascicles by the width of the distal fascicles.

## Dissociated motoneuron culture and in situ zymography assay

For dissociated motoneuron culture, left and right cervical ventral spinal cord tissues were dissected from E12.5 OF1 or *Rfx3* pregnant mice. Neurons were dissociated and cultured as described previously (*Charoy et al., 2012*; *Cohen et al., 2005*). After 24 hr in vitro, neurons were incubated for 10 min at 37°C with DQ-Gelatin 20 µg/mL (Invitrogen, France). Cells were washed twice with warm PBS and fixed in 4% paraformaldehyde both containing 25 µM of GM6001 MMP inhibitor (Millipore, France-CC1100). The cultures were incubated with Islet 1/2 antibody (1/50; DSHB, Iowa, USA – 39.4D5) overnight at 4°C then for 2 hr with α-mouse Alexa-555 (1/400; Invitrogen, France) to detect motoneurons. Nuclei were counter-stained with bisbenzimide (Promega). For quantification, the number of cells expressing Islet1/2 with gelatinase activity is reported realtive to the total number of Islet-1/2-positive cells.

## Microarray analysis and quantitative real-time PCR

The GFP+ motor pool was laser-captured from E11.5 GFP+ mouse embryos from HB9::GFP transgenic mice frozen in −45°C isopentane. Captured tissues were lysed in the lysis buffer provided with the RNA purification kit (RNAeasy microkit, Qiagen, France ). RNA quality was assessed on an Agilent 2100 bioanalyser (Agilent Technologies, USA). L/R matching samples that had a RNA integrity number (RIN) above 9 were amplified (ExpressArt PICO mRNA amplification kit, Amp-tec-Exilone, France) and reverse transcribed (BioArray HighYield RNA Transcript Labeling, ENZO, France). cDNA quality was assessed on an Agilent 2100 bioanalyser before fragmentation and hybridization on an Affymetrix microarray (GeneChip Mouse 430 2.0, Affymetrix, ThermoFischer scientific, France). Expression normalizations and present or absent calls were performed in Affymetrix Expression Console Software. Fold change and filtering were performed in Excel. Transcripts were considered as being expressed if scored as present in at least one sample of each embryo. Transcripts were classified as differentially expressed if the fold change (FC) between left and right samples had the same trend for all embryos (same sign for log2 ratio) and was over 1.5 (FC > 0.58 or FC < −0.58 in log2) on average and for at least two of the embryos. Transcripts with very low expression (maximal normalized expression <200) were removed. Raw data are available on GEO under the accession number GSE84778.

Real-time PCR was performed using MIQE pre-validated *Mmp2* (qMmuCID0021124) and *Mnx1* (qMmuCED0040199) primers (BioRad, france). Data were normalized to *GAPDH* expression values (primers Fw: AGAACATCATCCCTGCATCC; Rv: ACACATTGGGGCTAGGAACA). Real time PCRs were performed in duplicate on amplified RNA prepared as described for the microarray. Laser-capture microdissection, RNA preparation, microarray and qPCR were performed at the ProfileXpert core facility (France).

## RNAscope in situ hybridization

RNAscope in situ hybridization (Advanced Cell Diagnostic, Ozyme, France) was performed on 14–20-µm cryosections according to the manufacturer's recommendations for fresh frozen samples, using *Mmp2* C3 and C1 proprietary probes (references 315937 and 315931-C3, ACD, Ozyme, France). Both probes gave the same pattern, which mirror the distribution observed using the conventional in situ procedure. *UBC* and *DapB* probes were used as positive and negative controls, respectively (references 310777 and 312037, ACD, Ozyme, France). All incubations were performed in the HyBez hybridation system (ACD, Ozyme, France). Sections were fixed in 4% paraformaldehyde for 15 min before dehydratation and incubated in pretreat buffer 4 (Advance Cell Diagnostic, Ozyme, France) for 15 min at room temperature. DAPI staining was performed at the end of the procedure. The left and right side of the cervical spinal cord were imaged at 20x on a FV1000 confocal microscope (Olympus, France) using the same acquisition parameters. Labeled surfaces were quantified in ImageJ in ROI drawn from the DAPI staining. The threshold calculated on

the sum of the Z-stack image of one side was applied to the other side. Surface ratios were calculated after normalization to the selection area.

## Statistical analyses

Control and mutant embryos were from the same litters. All analyzable samples (diaphragm, western blots, cells or explants) were included, no outliers were removed. Left and right samples were from the same embryos. Analyses of the diaphragm innervation and *Mmp2* quantitative in situ were performed blind. No blinding was done on other data collections or analyses. Sample sizes, statistical significance and tests are stated in each figure and figure legend. All statistical analyses were done using Biostat-TGV (CNRS). Mann-Whitney (method: Wilcoxon rank sum) or Wilcoxon signed rank were used for small-sized samples or when distributions were not normal. Wilcoxon signed rank was used when paired analysis was needed (left versus right from the same embryo).

# Acknowledgements

We gratefully acknowledge L Schaeffer, M Carl and V Bertrand for helpful discussions and M Tata for proofreading the manuscript. We thank S Croze and C Rey (ProfileXpert, Lyon, France) for Microarray and qRT-PCR analyses. We thank A Huber (Neuherberg, Germany) for the Hb9::GFP mouse line and A Chédotal (Paris, France) and M Tessier-Lavigne (New-York, USA) for the Robo1/Robo2 mouse lines. This work was performed within the framework of the Labex CORTEX and Labex DevWeCAN of the Université de Lyon, within the program 'Investissements d'Avenir' (ANR-11-IDEX-0007) operated by the French National Research Agency (ANR). CC was funded by a doctoral fellowship from the Fondation pour la Recherche Médicale (FRM FDT20130928169) and a postdoctoral fellowship from the International Brain Research Organisation (IBRO). DMM was supported by R01 DC009410.

# Additional information

## Funding

| Funder | Grant reference number | Author |
|---|---|---|
| Fondation pour la Recherche Médicale | FDT20130928196 | Camille Charoy |
| International Brain Research Organization | Post-doctoral fellowship | Camille Charoy |
| Labex | Labex Cortex and Labex DevWeCan, ANR-11-IDEX-0007 | Valerie Castellani |

The funders had no role in study design, data collection and interpretation, or the decision to submit the work for publication.

## Author contributions

CC, performed the experiments, analyzed the results and contributed to writing the manuscript; SD, YC, IS, MB, KK, performed the experiments and analyzed the results; LMor, BD, JMS, provided embryos and provided information for protocols; LDG, LMoo, provided embryos and information for protocols; MS, provided antibodies and information for protocols; CR, provided embryos and information for protocols and wrote the manuscript; JFM, provided embryos; DMM, provided embryos, provided information for protocols and contributed to writing of the manuscript; JF, performed the experiments, analyzed the results, designed the experimental plan and supervised the study and wrote the manuscript; VC, designed the experimental plan, supervised the study and wrote the manuscript

## Author ORCIDs

Julien Falk, http://orcid.org/0000-0001-8590-5615
Valerie Castellani, http://orcid.org/0000-0001-9623-9312

## Ethics

Animal experimentation: This work was conducted in accordance with ethical rules of the European community and French ethical guidelines.

## Additional files

### Supplementary files

• Supplementary file 1. MMP2 addition on cervical ventral spinal cord increases Robo1 short form. This file provides the statistical report and individual values used to calculate the fold change presented in the discussion.

• Supplementary file 2. Robo1 short form ratio in Mmp2$^{-/-}$ cervical spinal cord. This file provides the statistical report and individual values for the fold changes presented in the discussion.

### Major datasets

The following dataset was generated:

| Author(s) | Year | Dataset title | Dataset URL | Database, license, and accessibility information |
|---|---|---|---|---|
| Julien Falk, Muriel Bozon, Valerie Castellani, Catherine Rey, Séverine Croze, Joel Lachuer | 2017 | Comparaison of left and right cervical motoneurons transcriptome | http://www.ncbi.nlm.nih.gov/geo/query/acc.cgi?acc=GSE84778 | Publicly available at the NCBI Gene Expression Omnibus (accession no: GSE84778) |

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
