## [Decision Letter]

Thank you for submitting your article "Genetic specification of left-right asymmetry in the diaphragm muscles and their motor innervation" for consideration by *eLife*. Your article has been reviewed by three peer reviewers, one of whom, Carol A Mason (Reviewer #2), is a member of our Board of Reviewing Editors, and the evaluation has been overseen by a Reviewing Editor and K VijayRaghavan as the Senior Editor.

The reviewers have discussed the reviews with one another and find the study of great value. The Reviewing Editor has drafted this decision to help you prepare a revised submission.

Summary:

Charoy et al. address the question of the development of left-right asymmetry of motor neuron innervation patterns in the phrenic nerves innervating the diaphragm, with the right nerve exhibiting longer and more extensive branching compared to the left. Left-right asymmetric patterning in the embryo as a whole has been linked to early morphogen gradient asymmetry, and left-right asymmetry of individual neuronal cell identity has been analyzed in *C. elegans* but how such asymmetry is translated into asymmetric innervation patterns is poorly understood.

They first use two genetic models deficient for factors controlling Nodal signaling and that impact asymmetry of visceral organs, and find that the patterning of diaphragm muscles or organs that constitute intermediate targets of phrenic motor neurons do not obviously influence L-R phrenic nerve differences. The authors argue that instead this could be due to a distinct genetic program operating within phrenic motor neurons on each side of the spinal cord. Explants containing phrenic motor neurons display differential outgrowth patterns, and together with the preceding genetic manipulations lead the authors to conclude that there must be intrinsic L-R molecular differences inherent to phrenic motor neurons.

They confirm these L-R molecular differences by expression profiling experiments that reveal L-R differentially expressed genes. Several of these genes are involved in chromatin modification and RNA processing, but these were not pursued except as evidence that there is L/R asymmetry in intrinsic "imprinting" of phrenic motor neurons. They go on to focus on Robo-Slit signaling because of previous published work (Tessier-Lavigne lab) and nicely show that while there was no lateralized expression of Robo1 transcript, there is localized expression of the short form of Robo1 in the right population of motor neurons. Associated with this post-translational modification are *MMP2* transcripts enriched in the right hand population, as was metalloprotease activity measured by other means. Importantly, in line with these conclusions, mutation of Robo1, *MMP2* and Abl1, a Robo1 effector, in mice appears to alter the outgrowth and/or branching pattern of the phrenic nerves such that their left-right asymmetry is reduced.

Thus, they conclude that a 1.15 fold enrichment of the short Robo1 isoform in R phrenic motor neurons is the main determinant of L-R asymmetry of phrenic innervation pattern. Overall, the study makes a compelling case that there are indeed gene expression differences within motor neurons on the left and right sides of the spinal cord that impact the asymmetry of phrenic axon outgrowth and branching patterns.

These findings are novel and interesting, as they provide some of the first mechanistic insights into this phenomenon. The manuscript is well supported by the data, which are convincing and clear, and the study has relevance for numerous labs in the field revisiting pre-target organization in tracts.

Nevertheless, there are a number of major concerns that need to be addressed, and several minor points that can be addressed textually. The suggested revisions are easily doable considering the authors have the mice at hand, and some of these deal with already generated data.

Essential revisions:

1) Addition of supplemental imaging controls to demonstrate the reproducibility and quantification schemes used to assess L-R asymmetry of the phrenic nerves across animals:

a) Some of the left-right traces do not incorporate all of the projection data, which may inflate the matches/discrepancies in the overlay of L vs R nerves. Showing sample variability would strengthen the rigor of the analyses.

b) The methods used to score the branching patterns of phrenic axons are not consistent. The red-green trace projections lack branches and details that are visible in the primary data images (e.g., the control nerves in with Figure 4G). It would be helpful to see more examples of the raw and "traced" data that were used to generate the data plots, as supplemental data. One suggestion for appreciating the stereotypy of projections of left and right nerves from embryo to embryo is to overlay the traces from multiple specimens. This would help readers assess the background level of deviation from nerve to nerve.

2) Mechanisms of how enrichment of the short isoform of Robo 1 specifies a different phrenic nerve pattern on the right side:

a) An important further inquiry, which could be done with the mice in hand, is to determine whether cleavage of Robo1 on the left vs. right populations of motor neurons altered in *MMP2* mutants. This would bolster the conclusion that differential levels of *MMP2* are crucial for setting the L-R differences in Robo1 processing. As they currently stand, the data connecting *MMP2* to Robo1 is based solely on the *MMP2* expression pattern and gain of function experiments, which are suggestive yet not definitive.

b) The authors should more directly state how they envision the signaling to work. Is asymmetry, as presented in the quantification, a reflection of left-right differences in branching and outgrowth? Robo1 has previously been implicating in branching/outgrowth. They might comment on how their results compare to data in "Biochemical purification of a mammalian slit protein as a positive regulator of sensory axon elongation and branching", Cell 96, 771-784 (1999). This study is in line with the descriptive observations in the present manuscript.

c) Comment on the fact that there seem to be no L-R Robo1 ligand differences.

3) One of the interesting aspects is analysis of differences in fasciculation and growth behavior of phrenic motor neurons; unfortunately, these data are relegated to Figure 3—figure supplement 1. The methods are given only in the Figure legend and not at all in the Methods for the Supplementary Figures but are quite unintelligible. The differences seem to be great, and although it is unclear whether Robo-Slit signaling would be involved in fasciculation as proposed by Jaworski and Tessier-Lavigne in their earlier study, these data are of interest given the L/R asymmetry issue and are therefore important data for the present study. Even if the data stay in Supplementary data, issues include: why do axons extend only from one aspect of the explant? In f. why was only one aspect of the explant measured for bundle width, and what is the genotype of the explant? Also statistical analysis is not well stated for this aspect of the study.

4) There is a noticeable gap in connecting the very apt initial observations of changes in phrenic nerve asymmetry in Pitx2 and Rfx3 mutants to the later findings of changes in motor neuron expression of *MMP2*, Robo1 processing, etc. Assessment of changes in the L-R differences in *MMP2* and Robo1 processing in Pitx2 and Rfx3 mutants would improve the cohesion of the studies. If *MMP2*/Robo1 processing are involved, there should be an equilibration of *MMP2*/Robo1 levels in left vs right phrenic motor neurons in Pitx2 and Rfx3 mutants.

[Editors' note: further revisions were requested, as described below.]

Thank you for submitting your work entitled "Genetic specification of left-right asymmetry in the diaphragm muscles and their motor innervation" for consideration by *eLife*. Your article has been reviewed by two peer reviewers, and the evaluation has been overseen by a Reviewing Editor and a Senior Editor. The following individuals involved in review of your submission have agreed to reveal their identity: Bennett G Novitch.

As you can see in the attached reviews, both reviewers found the requested revisions of the original manuscript satisfactory. Reviewer 2 (called reviewer 5) indicates that the revision is a more balanced presentation of results supporting the conclusions that L-R asymmetry in innervation of the diaphragm is due to intrinsic differences in phrenic motor neurons and subtle differences in Slit-Robo1 signaling. The reviewers also agree that the data and narrative are well presented, and the story is novel and interesting, as few studies have approached the question of laterality in mammalian motor innervation, and thus they considered the message of the study a valuable contribution to the field.

Nonetheless, reviewer 1 has pointed to a lack of statistical significance in experiments on the cleavage of Robo1 between L and R motor neuron explants and on the impact of *MMP2* treatment on the enrichment of the short Robo1 isoform. We appreciate that you have gone to great lengths in the resubmission to perform experiments on immunoblotting/inhibition of MMP activity, and dissection/enrichment of motorneurons and western blotting to detect the differential MMP levels that reflect a different pattern of Robo processing in the left and in the right cervical motoneuron population. However, after posting their reviews, the reviewers and Reviewing Editor engaged in much discussion (not included here): reviewer 2 saw that reviewer 1 identified the lack of statistical significance, then re-ran your statistical analyses and found that they indeed do not reach significance. Reviewer 2 also arrayed the data in different graphical formats such as columns/box and whiskers. These showed little difference in the mean beyond individual sample variability. This reviewer suggested that you might consider showing whiskers for these graphs, as it might more fairly represent whether or not there are L-R differences.

You state that *MMP2* activity is higher in Right motoneurons, accounting for Robo1 isoform enrichment, and cite statistics in the figure legends, but do not overtly cite the lack of significance in the text. Indeed, the results as they are written appear convincing but then appear tenuous when the statistics in the Figure legends are considered. The reviewers now conclude that any in vivo differences in Robo1 processing between L and R phrenic motor neurons may be very minor.

Thus, your hypothesis that cleavage of Robo1 to the short isoform is through *MMP2*, resulting in R-L differences in symmetry in phrenic nerve innervation of the diaphragm, is not robustly supported. To increase the n for analyses of mutant tissue, and to reach statistical significance, you indicate that you would have to "replicate these experiments at least 10 times, especially tricky as these experiments are performed on knockout tissue". This effort would indeed be long and arduous; we do not ask at this point that you do this simply to add more n's.

The reviewers, Reviewing Editor, and Senior Editors agree that valuable observations are presented in the first part in Figures 1–4 – on patterns of branching, that asymmetry is controlled by Nodal signaling and seen in "intrinsic" differences in phrenic motoneurons, with the short isoform of Robo enriched on the right side to presumably control fasciculation differently than on the left. These are data well worth publishing and have merit even if the mechanistic aspect is unclear. Ordinarily, as with many of the other prominent journals, for the manuscript is to be acceptable, the authors would need to additionally provide mechanism. We understand that not all studies can end with a definitive mechanism. In repeated consultation with the reviewers, the Reviewing Editor, and the Senior Editor, we initially considered recommending that you either omit the experiments that are not substantiated by the statistical analyses, or describe the results as evidence that your hypothesis was not supported by the data. In either case, this would diminish the offering of the study for *eLife*, and we imagine would not be agreeable to you. Thus, the end, we have decided that the paper with its current components and incomplete outcome of analyses is not acceptable for publication in *eLife*. Instead, we encourage that you rethink the study with sufficient statistical power to ensure that you will have adequate sample size for a strong conclusion on L-R differences in Robo1 processing through *MMP2*, the proposed route through which L-R axon asymmetry could be achieved, either way.

*Reviewer #1:*

In the revised manuscript *eLife*"Genetic specification of left-right asymmetry in the diaphragm muscles and their motor innervation", the authors extend their study by attempting to address some of the criticisms raised in the *eLife* consensus review. The authors address satisfactorily the criticisms raised in review point 1 re: quantification of defective L-R asymmetry of diaphragm innervation. However, the authors fail to address point 2 regarding the impact of *MMP2* levels on the enrichment of the short Robo1 isoform, and the mechanism of Robo1 specification of L-R phrenic asymmetry. Here the experiments comparing Robo cleavage between left and right side do not reveal a statistically significant difference (Figure 5B, p=0.187), and MMP inhibition does not result in significant changes in Robo1 cleavage asymmetry (Figure 5F). Importantly, as already shown in the original manuscript, the treatment of motor neuron explants with *MMP2* also fails to significantly change the ratio (Figure 5C, p=0.125).

Furthermore, since normal embryos do not show significant left-right differences in Robo1 cleavage (Figure 5B, p=0.187), it is difficult to interpret the observation that explants from *MMP2* mutant embryos do not display significant changes in L-R differences in Robo1 cleavage abundance (Figure 5—figure supplement 1K p=0.25). I am also confused about this experiment since in the rebuttal letter the authors mention that they examine "2 independent littermates of 5 embryos", compared to wild types, yet in the data table they mention an n of 3 replicates. The comment about the difficulty of extending these to a robust number of replicates makes me doubt whether additional experiments are feasible. Although other major comments are addressed in a satisfactory manner, the critical link between *MMP2*, Robo1 cleavage and diaphragm innervation asymmetry remains tenuous.

*Reviewer #2:*

The revision has addressed most of my concerns, and presents a much more balanced presentation of results supporting the conclusions that L-R asymmetry in diaphragmatic innervation is due to intrinsic differences in phrenic motor neurons including subtle differences in Slit-Robo1 signaling. The data and narrative are well presented, and the story as a whole is novel and interesting. Few studies have approached the question of laterality in mammalian motor innervation, so it is a valuable contribution to the field.

[Editors' note: further revisions were requested, as described below.]

Thank you for resubmitting your work entitled "Genetic specification of left-right asymmetry in the diaphragm muscles and their motor innervation" for further consideration at *eLife*. Your revised article has been favorably evaluated by K VijayRaghavan (Senior editor), a Reviewing editor, and two reviewers.

The reviewers, both new, feel that your presentation in this third submission is much improved over the previous two versions, which they have viewed. The reviewers thought your findings on left-right asymmetry in phrenic nerve innervation of the mouse diaphragm are important, particularly the nodal-mediated signaling of the asymmetry, the anatomical differences in the fasciculation differences in vitro, and the molecular screen pinpointing MMP processing of the Robo receptor to effect the l-r differences. However, there are several issues that need to be addressed before acceptance, as outlined below and in the appended reviews:

In your last rebuttal and in the text, you explain two major amendments: First, you state that you addressed the lack of statistical significance in the L/R difference in Robo forms in immunoblots, and we acknowledge your efforts. However, one of the present reviewers criticized the western blot analysis with the new statistical test used (based on the Degaspari method) and considered the normalization strategy invalid. Upon normalizing the levels of the short band to the full length band, one would predict that if the short form derives from the long form, the levels of the long form would decrease and the short form would increase. Please comment.

Second, your previous analysis of Robo processing by *MMP2* in spinal cord tissue did not reach statistical significance. As with the previous reviewers, the present reviewers stressed that analyzing the mutants is a key experiment, and that it is crucial to your argument as such data would demonstrate a biochemical link between *MMP2* activity and Robo1 receptor cleavage. Increasing the "n" would strengthen your case, but in a potentially "dangerous' direction, one of seeking the desired result and this route would be a lengthy as well.

You observed a partial symmetrization of the phrenic branches, with a right pattern that resembles the left pattern in control littermates in E14.5 *Mmp2*-/- embryos. But now you take the stance that the defect observed with *MMP2* genetic deletion might affect alternative, or additional signaling pathways in axon guidance and that *MMP2* may be one of several proteases, and thus chose to present the findings on the Robo and *MMP2* pathways in phrenic nerve left-right asymmetries "independently" from each other, suggesting but not concluding that this relationship may be linked. While this does not solidify "mechanism", given the rounds of work you have done to improve the manuscript, we now feel that your revision is acceptable. However, we request that you acknowledge overtly that your attempts to demonstrate Robo1 processing in *MMP2* mutants lacked significance, and shorten the text in the Discussion where you discuss multiple *MMP2* targets and proteases.

In addition to these two major points of contention by the various the reviewers, as often happens with re-review by a new set of reviewers, additional experiments were suggested. One suggestion is to determine whether the cleaved Robo1 receptors have any guidance or fasciculation function. Your explant system would be ideal to investigate how *MMP2* mutant motor neurons respond to Slits compared to wild type neurons after bath application or by co-culture with Slit-expressing cells, an analysis that would strengthen your hypothesis and data on the asymmetric axon innervation patterns in vivo. However, this would also take time and hopefully could be the basis of "next-step" experiments.

*Reviewer #3:*

I have carefully read all the different versions of the manuscript. I believe the authors have made a huge effort to improve the manuscript along the revision period. The last version is a compelling piece of work that, through many different approaches, demonstrates that right and left phrenic motor neurons have different intrinsic genetic profiles. In general, the way they present these data in the last version is much better and clear than in previous versions and, in my opinion, this manuscript deserves to be published in *eLife*.

However, I also believe that there is an important issue raised by previous reviewers that has not been yet addressed in the last version: The detection of the different forms of Robo1 by WB in the L/R sides of *MMP2* mutants. In the second point-to-point letter to the reviewers (point 2) the authors state they attempted to address this issue by conducting a pilot experiment using 2 littermates of 5 embryos each and found a reduction in the L/R ratio in the *MMP2* mutants compared to the controls, but the difference was not significant. They then stressed that, given the nature of the samples, it would be necessary to pool a higher number of embryos to reach statistical significance and that, constraints in the time of revision limited their capability to get the necessary amount of tissue. I believe this is a key experiment they need to complete no matter how long it takes to get the tissue, specially since they got promising results in the pilot experiment.

*Reviewer #4:*

Overall, this paper makes several important findings, but a set of major concerns suggest a reanalysis is needed of the Robo1 western blot experiment, and that a new set of experiments are needed for the axon explants. Key strengths of the study include interesting results that document a left-right asymmetry in phrenic nerve innervation of the embryonic mouse diaphragm, which would be a new system for left-right asymmetries in brain development. Their evidence indicates that Nodal signaling mediators are required for this asymmetry. A particularly significant finding that left and right motor axons have differences in their in vitro outgrowth behavior (area covered, fasciculation), as this suggests that intrinsic left-right specification, and a molecular screen has identified a large number of molecular differences, including asymmetries in MMP expression and activity. Also interesting is that mutations in the Robo receptors causes a switch to symmetrical innervation in the diaphragm, which suggests a molecular mechanism to explain at least part of the innervation asymmetries.

A key but problematic set of evidence is a potential asymmetry in Robo1 protein, which appears to derive from increased level of cleavage of Robo1 protein on the right side of the spinal cord. This result is important because it would be significant to find asymmetry in receptor processing, and to correlate this with altered axon pathfinding or fasciculation. However, several concerns arise about these experiments.

a) The material comes from dissected halves of the spinal cord, tissue lysates followed by Western blotting. However, Robo1 expression is widespread among many neuron types in the spinal cord, and so is not specific for motor neurons. So, while an asymmetry in Robo1 cleavage could be consistent with the motor neuron pathfinding differences, this result is limited in that it does not show where the cleaved receptors are. This is not a fault of the authors, because tools to map the distribution of cleaved Robo1 protein in tissue are not available. This point should be discussed as a caveat.

b) A related concern is that whether or not Robo1 receptor proteins are cleaved in the spinal cord does not provide any evidence about Robo cleavage in the site of Robo1 action in the peripheral axons. Again, this seems impossible to assess currently, but should be discussed as a caveat.

c) The Robo1 Western blot experiment has been revised to normalize the levels of the short band to the full length band, according to Figure 5B label on the second graph. This normalization strategy does not seem valid, because the short form is hypothesized to be derived from the long form. This leads to the prediction that the right side increase in the short form should accompanied by a decrease in full length. If so, then the normalization strategy would inappropriately exaggerate the increase in the short form. The paper cited for normalization strategies (Despagari 2014) does not provide an obvious justification for this short/long normalization, either. Therefore, serious doubts are raised about the validity of this important claim, which undercuts a major strength.

d) It is not clear whether cleaved Robo1 receptors have altered or indeed any guidance or fasciculation function in this system, such as decreased Slit responses or increased fasciculation. With the explant outgrowth system, it should be within the expertise of the investigators to directly test responses of motor neurons to Slits applied in the culture bath or by co-culture with Slit-expressing cells. This new experiment would significantly strengthen the paper by testing the prediction of altered axon responses in culture that might correlate with the in vivo axon projection patterns.

---

## [Author Response]

*Essential revisions:*

*1) Addition of supplemental imaging controls to demonstrate the reproducibility and quantification schemes used to assess L-R asymmetry of the phrenic nerves across animals:a) Some of the left-right traces do not incorporate all of the projection data, which may inflate the matches/discrepancies in the overlay of L vs R nerves. Showing sample variability would strengthen the rigor of the analyses.*

*b) The methods used to score the branching patterns of phrenic axons are not consistent. The red-green trace projections lack branches and details that are visible in the primary data images (e.g., the control nerves in with Figure 4G). It would be helpful to see more examples of the raw and "traced" data that were used to generate the data plots, as supplemental data. One suggestion for appreciating the stereotypy of projections of left and right nerves from embryo to embryo is to overlay the traces from multiple specimens. This would help readers assess the background level of deviation from nerve to nerve.*

We fully understand these concerns. It is effectively a major point to make it clear to the referees and the readers that all our quantifications were done on immunolabeled diaphragms, and not based on the traces that we showed in the pictures. Indeed, the traces were thought to help the reader identifying the branches of interest and to better appreciate the general phrenic nerve pattern of the primary and secondary order branches that innervate the lateral muscles. We thought that simplified traces also nicely highlight the different angles by which left and right primary nerves split into dorsal and ventral branches (T and V shapes).

In the revised version, we rearranged the Figure 1 and its supplements to provide additional illustrations to clarify these points.

To better explain which branches were considered, we included an image of the color-coded traces of lateral and crural branches drawn from Neurofilament staining of wholemount diaphragm (Figure 1—figure supplement 1A). We also added an image describing the method used for quantification in Figure 1—figure supplement 1B.

To demonstrate the stereotypy of the left-right asymmetry that we measured, we incorporated in Figure 1D-E the histograms showing the average value measured for L and R phrenic nerves together with the SEM, which illustrates the variation observed between individual embryos for the 2 parameters measured. In addition, we added in Figure 1—figure supplement 3B a ladder graph showing the paired left and right values of the defasciculation distance for eight E14.5 embryos. As suggested, we now provide several examples of wholemount diaphragms labeled with neurofilament with their respective branch traces (Figure 1—figure supplement 3C). We hope this will better document the stereotypy of the asymmetry.

*2) Mechanisms of how enrichment of the short isoform of Robo 1 specifies a different phrenic nerve pattern on the right side:*

*a) An important further inquiry, which could be done with the mice in hand, is to determine whether cleavage of Robo1 on the left vs. right populations of motor neurons altered in MMP2 mutants. This would bolster the conclusion that differential levels of MMP2 are crucial for setting the L-R differences in Robo1 processing. As they currently stand, the data connecting MMP2 to Robo1 is based solely on the MMP2 expression pattern and gain of function experiments, which are suggestive yet not definitive.*

We fully understand the relevance of this concern and agree on the fact that we did not provide a direct link between Robo protein forms and *MMP2*. We worked in two directions to strengthen our conclusions. We first thought that a quite feasible way to document this link would be to show that manipulating endogenous MMP activity in the Left and Right motoneuron fraction impacts on Robo processing. To do so, we dissected cervical ventral spinal cord tissue enriched in phrenic motoneurons from HB9+ embryos. First Left and Right tissues were let for two hours in zymography buffer which preserves MMP activity and then processed for immunoblot to compare Robo protein forms. First, we found that the Right/Left ratio of Robo cleaved form was maintained during the incubation, further supporting that higher MMP activity in the Right population results in higher rate of Robo processing. Second, we inhibited endogenous MMP activity in the motoneuron tissues using GM6001. Under this condition, short forms of Robo1 are not anymore Right-enriched. We hope that these experiments further show that differential MMP activity can generate a different pattern of Robo processing in the left and in the right cervical motoneuron population.

The second direction, suggested by the referees, was to examine whether the L/R ratio of Robo processing is modified in *MMP2* mutants. Although very pertinent, we anticipated that it would be highly challenging to provide significant conclusion, using western blot approach, which is the only way for distinguishing long and short Robo forms. First, the amplitude of the L/R difference is narrow. Second MMP mutants are reported to show high levels of compensation (Holmbeck et al., 2004; Kukreja et al., 2015) and we have observed during the analysis that levels of phrenic nerve symmetrization in *MMP2* mutants is variable. Nevertheless, we conducted these experiments and crossed *MMP2* and HB9+ lines to highlight the motoneuron population for our dissection procedure. Two independent littermates of 5 embryos each were analyzed and L/R ratio was reduced, compared to wild- types. This analysis thus provides additional support for linking MMP activity and Robo processing. We would like to stress that given the nature of the samples, it is necessary to pool a substantial number of embryos. Reaching statistical significance would mean replicating these experiments at least 10 times, which was not feasible during the revision timing and indeed incompatible with the fact that these experiments are performed on knockout tissue.

Nevertheless all together, we are convinced that these additional results strengthen our interpretation and hope they now provide convincing support for contribution of *MMP2* to the differential processing of Robo1 in left and right motoneurons.

*b) The authors should more directly state how they envision the signaling to work. Is asymmetry, as presented in the quantification, a reflection of left-right differences in branching and outgrowth? Robo1 has previously been implicating in branching/outgrowth. They might comment on how their results compare to data in "Biochemical purification of a mammalian slit protein as a positive regulator of sensory axon elongation and branching", Cell 96, 771-784 (1999). This study is in line with the descriptive observations in the present manuscript.*

We agree that several properties have been assigned to Slits, which need to be considered in our interpretation of the Slit/Robo function in the setting of Left and Right phrenic innervation and are now discussed in the revised manuscript (Discussion section). Indeed, when reaching the diaphragm, the phrenic nerves undergo successive defasciculations, leading to the individualization of axonal bundles, referred to as “branches”. This term gives confusion because these bundles are not “branches per see” meaning produced by sprouting of novel axon from pre-existing one. In the terminal step of the innervation, individual axons undergo branching, but this is a stage that we did not investigate in our study, concentrating on the initial defasciculation processes. Therefore, we believe that the asymmetry of phrenic patterns reported here result from differential regulation of axon defasciculation and growth, rather than branching per se. This interpretation is coherent with the study by Jaworski and Tessier-Lavigne (Jaworski and Tessier-Lavigne, 2012), reporting that defasciculation is affected by loss of Slit-Robo function, both receptors and ligands being produced by the motoneurons themselves. The simplest mechanistic model would be that Slit-Robo interactions between phrenic axons promote fasciculation. This signaling is higher in the Left than in the Right population, due to higher Abl activity and lower Robo processing in the Left than in the Right. We do not exclude that the Slit-Robo signaling could also regulate other aspects, such as axon speed, orientation of growth in the target, and terminal branching, but studying these contributions were the main focus of the present study.

*c). Comment on the fact that there seem to be no L-R Robo1 ligand differences.*

We now introduced a section of discussion in our manuscript, in which we quote that an asymmetric signaling can be generated with equivalent initial levels of Slits.

*3) One of the interesting aspects is analysis of differences in fasciculation and growth behavior of phrenic motor neurons; unfortunately, these data are relegated to Figure 3—figure supplement 1. The methods are given only in the Figure legend and not at all in the Methods for the Supplementary Figures but are quite unintelligible. The differences seem to be great, and although it is unclear whether Robo-Slit signaling would be involved in fasciculation as proposed by Jaworski and Tessier-Lavigne in their earlier study, these data are of interest given the L/R asymmetry issue and are therefore important data for the present study.*

Additional information has been added in the method section, in a separate paragraph, so the reader can easily have access to our method of quantification.

*Even if the data stay in Supplementary data, issues include: why do axons extend only from one aspect of the explant? In f. why was only one aspect of the explant measured for bundle width, and what is the genotype of the explant? Also statistical analysis is not well stated for this aspect of the study.*

Our figure was unclear since it induced confusions. Indeed, quantifications have been made on the whole explants, without any bias of choices. The magnified panels showing only part of the explants were presented to more clearly illustrate the different fasciculation aspect, which was not visible in pictures of whole explants. We now clarified the figure legend and stated in the experimental procedure that measures were done all around the explants. The statistics have been inserted in the corresponding figure legend (and the transparent form).

*4) There is a noticeable gap in connecting the very apt initial observations of changes in phrenic nerve asymmetry in Pitx2 and Rfx3 mutants to the later findings of changes in motor neuron expression of MMP2, Robo1 processing, etc. Assessment of changes in the L-R differences in MMP2 and Robo1 processing in Pitx2 and Rfx3 mutants would improve the cohesion of the studies. If MMP2/Robo1 processing are involved, there should be an equilibration of MMP2/Robo1 levels in left vs right phrenic motor neurons in Pitx2 and Rfx3 mutants.*

Unfortunately, we have no access anymore to the Pitx2 colony. Fixed embryos were provided by our collaborator, Dr D. Martin, who did not maintain the colony in her lab. Nevertheless, to address this point, we concentrated on RFX3 mutant embryos. We had to consider several constraints. First, due to the partial penetrance of the phenotype, embryos must be considered individually. Second, we need to focus on MMP activity in motoneurons, excluding MMP activity in other populations of the ventral spinal cord, such as the progenitors that express it at high levels. This excludes the approaches of western blot and RTPCR because we cannot separate progenitors and motoneurons in the dissection procedure.

We therefore selected the technique of in situ zymography, that is very sensitive, in dissociated motoneuron cultures, that allow assessing MMP activity of motoneurons isolated from embryos that we previously identified as affected, based on their lung isomerism. The analysis demonstrated that L/R difference of MMP activity is gone in these embryos. These data have been inserted in Figure 5E.

These new data thus establish a link between early L/R specification and the molecular effectors of L/R asymmetry in phrenic motoneurons, which we hope meets the expectation of the referees.

[Editors' note: further revisions were requested, as described below.]

*[…] The reviewers, Reviewing Editor, and Senior Editors agree that valuable observations are presented in the first part in Figures 1–4 – on patterns of branching, that asymmetry is controlled by Nodal signaling and seen in "intrinsic" differences in phrenic motoneurons, with the short isoform of Robo enriched on the right side to presumably control fasciculation differently than on the left. These are data well worth publishing and have merit even if the mechanistic aspect is unclear. Ordinarily, as with many of the other prominent journals, for the manuscript is to be acceptable, the authors would need to additionally provide mechanism. We understand that not all studies can end with a definitive mechanism. In repeated consultation with the reviewers, the Reviewing Editor, and the Senior Editor, we initially considered recommending that you either omit the experiments that are not substantiated by the statistical analyses, or describe the results as evidence that your hypothesis was not supported by the data. In either case, this would diminish the offering of the study for eLife, and we imagine would not be agreeable to you. Thus, the end, we have decided that the paper with its current components and incomplete outcome of analyses is not acceptable for publication in eLife. Instead, we encourage that you rethink the study with sufficient statistical power to ensure that you will have adequate sample size for a strong conclusion on L-R differences in Robo1 processing through MMP2, the proposed route through which L-R axon asymmetry could be achieved, either way.*

Two major issues were raised by the referees: i) lack of statistical significance for the L/R difference in Robo forms observed in immunoblots of motoneuron-containing tissue, and ii) lack of statistical significance for the analysis of Robo processing by *MMP2*. We have addressed these concerns as follows.

i) Lack of statistical significance for the L/R difference in Robo forms observed in immunoblots of motoneuron-containing tissue.

The data presented in our prior manuscript were not statistically significant and therefore did not support our conclusion of a L/R difference in Robo forms in spinal cord motor neurons. We have now recognized that this was due to an inadequate method of analysis. To address this issue, and as recommended in the literature (Degasperi et al., 2014), we have now normalized the L/R data for comparison of L/R data between different western-blots and obtained statistical significance. The source data file provided with the new version of the manuscript contains the raw and normalized data. To consider that data obtained by immunoblotting may not follow the normal distribution assumption required for the t-test that is classically used (Kreutz et al., 2007), we also performed a Mann-Whitney test. Both tests yielded similar and significant p values (p= 0.016200 t-test; p=0.015873 Mann-Whithey). Thus, we believe that the statistical analysis of our data now supports the conclusion that Robo forms differ in left and right motoneuron-containing spinal cord tissue.

ii) Lack of statistical significance for the analysis of Robo processing by *MMP2*.

Even though MMP can process Robo (Seki et al., 2010), our analysis of Robo processing by *MMP2* in spinal cord tissue did not reach statistical significance. This may have several reasons, including the likely possibility that *MMP2* is not the only protease involved in generating left-right asymmetries of Robo forms in spinal cord tissue. Accordingly, our microarray analysis showed that the genes for 7 MMPs and 13 ADAMs metalloproteinases are expressed in left and right motoneuron-containing spinal cord tissue. This includes Adam10, whose homolog in *Drosophila*, Kusbanian, has been implicated in Robo processing (Coleman et al., 2010). Moreover, different MMPs can cleave the same protein (Kukreja et al., 2015; Prudova et al., 2010) and some of the short Robo form is preserved in *MMP2* mutant tissue.

Finally, we had not previously explained the consideration that *MMP2* has several known targets in addition to Robos (Dean and Overall, 2007), and several of these are also expressed by phrenic motoneurons and involved in their patterning, such as NCAM (Allan and Greer, 1998; Dean and Overall, 2007). Accordingly, the defect observed after *MMP2* genetic deletion might affect alternative, or additional signaling pathways involved in axon guidance.

For this reason, we have revised our manuscript to present the findings on the Robo and *MMP2* pathways in phrenic nerve left-right asymmetries independently from each other and only discussed a possible link. Accordingly, we have presented two examples of signaling pathways that contribute to the generation of left-right asymmetries in motor neurons. This revised format should better reflect the complexity of the signaling mechanisms underlying the establishment of left and right phrenic nerve patterns and whilst opening up the field to further investigation into the mechanisms underlying Robo processing (Blockus and Chédotal, 2016).

In summary, we have revised our manuscript substantially to underscore how a left-right genetic program initiated by Nodal sets into motion a complex signaling network that controls a range of events required for the stereotypic and asymmetric pattern diaphragm innervation.

List of major changes in the revised manuscript:

1) Improved statistical analysis of left and right differences in Robo forms detected by immunoblotting, including corresponding revision of the results and experimental procedures as well as data representation (Figure 5B).

2) Differences in the MMP and Robo pathways are now presented independently, and the wording that bridges the Robo and *MMP2* studies has accordingly been modified.

3) The Discussion has been modified in several ways. Firstly, the paragraph on Slit/Robo signaling was changed to better discuss potential roles of Robo1 asymmetric processing. Secondly, a new section has been added to discuss the role of *MMP2*, including its several substrates, and to explain why *MMP2* genetic loss is expected to lead to only a partial symmetrisation.

4) The analysis of ABL1 levels and Abl1 mutant phenotype has been removed, as ABL1 is not a mediator only of the Slit-Robo pathway, but operates downstream of several signaling receptors.

[Editors' note: further revisions were requested, as described below.]

*The reviewers, both new, feel that your presentation in this third submission is much improved over the previous two versions, which they have viewed. The reviewers thought your findings on left-right asymmetry in phrenic nerve innervation of the mouse diaphragm are important, particularly the nodal-mediated signaling of the asymmetry, the anatomical differences in the fasciculation differences in vitro, and the molecular screen pinpointing MMP processing of the Robo receptor to effect the l-r differences. However, there are several issues that need to be addressed before acceptance, as outlined below and in the appended reviews:*

*In your last rebuttal and in the text, you explain two major amendments: First, you state that you addressed the lack of statistical significance in the L/R difference in Robo forms in immunoblots, and we acknowledge your efforts. However, one of the present reviewers criticized the western blot analysis with the new statistical test used (based on the Degaspari method) and considered the normalization strategy invalid. Upon normalizing the levels of the short band to the full length band, one would predict that if the short form derives from the long form, the levels of the long form would decrease and the short form would increase. Please comment.*

We understand the concern of the referee. A normalization is obligatory to eliminate the experimental variability. Normalization to the full-length form is classically used for assessing protein processing (Gatto et al., 2014; Lin et al., 2008). Nevertheless, another option is to normalize to the tubulin. We obtained the same ratio when the normalization is done using the tubulin. Therefore, the variability of the long form between lines is mainly due to differences of loading. We thus kept in the principal figure the normalization on the protein of interest, and present the normalization to tubulin in the associated supplemental figure (Figure 5—figure supplement 1D).

More generally we agree with the referee that the short Robo forms might derive from the long Robo form. Nevertheless, we would like to highlight that such linear relationship between long and short forms cannot be deduced from western blot analysis. Indeed, it is not known whether the short and long forms have the same stability and half-life. Moreover, it is not known whether the Robo1 antibody recognizes with the same efficiency the long and the short forms.

Here are some additional elements on the western blot normalization. We normalized the lines within for each blot to take into account the variability of samples and deposition. This normalization enable comparison between left and right samples, but the normalized values vary between different Western blots for several reasons, which explains why a normalization between different blots is needed. First each western blot represents about 7 different embryos, individually dissected. This might generate some heterogeneity in the biological samples. Second, the re-use of antibodies and reagents to reveal the proteins might induce differences of band intensity between different western blots. Normalizing the western blots do not change the left-right ratio but allowed taking into account in the determination of the variability the heterogeneity between samples and reagents.

*Second, your previous analysis of Robo processing by MMP2 in spinal cord tissue did not reach statistical significance. As with the previous reviewers, the present reviewers stressed that analyzing the mutants is a key experiment, and that it is crucial to your argument as such data would demonstrate a biochemical link between MMP2 activity and Robo1 receptor cleavage. Increasing the "n" would strengthen your case, but in a potentially "dangerous' direction, one of seeking the desired result and this route would be a lengthy as well.*

*You observed a partial symmetrization of the phrenic branches, with a right pattern that resembles the left pattern in control littermates in E14.5 Mmp2-/- embryos. But now you take the stance that the defect observed with MMP2 genetic deletion might affect alternative, or additional signaling pathways in axon guidance and that MMP2 may be one of several proteases, and thus chose to present the findings on the Robo and MMP2 pathways in phrenic nerve left-right asymmetries "independently" from each other, suggesting but not concluding that this relationship may be linked. While this does not solidify "mechanism", given the rounds of work you have done to improve the manuscript, we now feel that your revision is acceptable. However, we request that you acknowledge overtly that your attempts to demonstrate Robo1 processing in MMP2 mutants lacked significance, and shorten the text in the Discussion where you discuss multiple MMP2 targets and proteases.*

We understand that the referees wish that we mention in the manuscript our attempt to link *MMP2* and Robo1 and the lack of statistical significance of the data. We have now added in the Discussion a paragraph stating:

– First that incubation of cervical spinal cord tissue with active *MMP2* significantly increased the short Robo1 forms (using the same normalization method as for assessing the left-right differences) (Supplementary File 1)

– And second that however, our analysis of Robo1 in cervical motoneuron tissue from *Mmp2* mutants did not allow to detect a significant difference. (Supplementary File 2)

We also shortened our Discussion on the contribution of other protease pathways.

*In addition to these two major points of contention by the various the reviewers, as often happens with re-review by a new set of reviewers, additional experiments were suggested. One suggestion is to determine whether the cleaved Robo1 receptors have any guidance or fasciculation function. Your explant system would be ideal to investigate how MMP2 mutant motor neurons respond to Slits compared to wild type neurons after bath application or by co-culture with Slit-expressing cells, an analysis that would strengthen your hypothesis and data on the asymmetric axon innervation patterns* in vivo*. However, this would also take time and hopefully could be the basis of "next-step" experiments.*

We fully agree that investigating the functional properties of the balance of short and long Robo forms is an important step towards understanding the Slit-Robo signaling. Effectively this is tricky because *MMP2* loss of function does not produce an optimal outcome on the balance. We have no entry point, neither from our own data nor from the literature, into how the Robo forms are produced and regulated, that would allow a straightforward investigation. This is also an additional reason for why this investigation certainly requires a fully dedicated investigation program.